# Learning Discrete Latent Models from Discrete Observations

## Abstract

A central challenge in machine learning is discovering meaningful representations of high-dimensional data, commonly referred to as representation learning. However, many existing methods lack a theoretical foundation, leading to unreliable representations and limited inferential capabilities. In approaches where certain uniqueness of representation is guaranteed, such as nonlinear *Independent Component Analysis*, variables are typically assumed to be continuous. While recent work has extended identifiability to binarized observed variables, no principled method has been developed for scenarios involving discrete latent variables. In this paper, we show how identifiability can be achieved when both latent and observed variables are discrete. We propose general identification conditions that do not depend on specific data distributional assumptions or parametric model forms. We further show how multi-domain information can be leveraged in this context to relax the constraints. The effectiveness of our approach is validated through experiments on both simulated and real-world datasets.

## 1 Introduction

Learning effective representations without supervision has always been critical to the performance of downstream deep learning tasks. In recent years, numerous advanced methods for representation learning have emerged, ranging from earlier models like Variational Autoencoders (VAEs) and Generative Adversarial Networks (GANs) to more recent innovations such as Diffusion Models Kingma (2013); Goodfellow et al. (2014); Sohl-Dickstein et al. (2015); Ho et al. (2020). These well-known unsupervised methods aim at learning an accurate posterior distribution over a lower-dimensional unobserved variable. It is hoped that by aligning observed distributions with the predicted ones, the learned posterior will correspond to the underlying distribution of statistically independent sources of variation. However, few of them have theoretical guarantees in terms of the learned representations, without which the results can be unreliable Arora et al. (2017); Dai & Wipf (2019). Given this limitation, recent developments in the field have focused on establishing the reliability of the learned representations by ensuring that they capture the true explanatory factors behind the observed data—a concept known as *identifiability*. With identifiability, we can guarantee that in the large sample limit, the probabilistic model learns a unique representation corresponding to the true latent factors. Identifiability is therefore crucial in representation learning, as it provides reliable interpretability, supports accurate inference, and enhances the usefulness of the learned representations for downstream tasks.

Among the representation learning methods that guarantee the identifiability of models, most focus on the linear setting. Well-known methods include *Independent Component Analysis (ICA)* Hyvärinen & Oja (2000), *Factor Analysis (FA)* Spearman (1904), *dictionary coding* Olshausen & Field (1997), and *latent class models* Goodman (1974). For instance, in *ICA*, the observed data is considered a mixture of independent, unobserved components, and the goal is to "demix" these observations to recover the latent variables, up to some ambiguity (please refer to Appendix A.1.1 for more details). Recent advancements have extended *ICA* to nonlinear settings, where deep neural networks are employed to approximate the nonlinear mixing process. Notable examples include iVAE (Identifiable Variational Autoencoders) and Invariant Causal Representation Learning Hyvärinen & Morioka (2019); Khemakhem et al. (2020); Schölkopf et al. (2021). Identifiability in nonparametric settings poses a significant challenge, as nonlinear transformations can obscure the original sources of variability. For instance, when variables are continuous, it is well-known that the model be-

comes severely unidentifiable if the observations are independent and identically distributed (i.i.d.) Hyvärinen & Pajunen (1999). To address this issue, additional constraints are typically required to ensure identifiability. One widely adopted approach is to leverage multi-domain information. Here, "multi-domain" refers to distinct scenarios or conditions under which the distributions of variables differ. This means that with the inclusion of domain information, represented by a fully observed random variable $\mathbf{u}$ (e.g., temporal or contextual information), we assume that each latent variable is statistically dependent on $\mathbf{u}$. Importantly, the mixing function that maps the latent variables to the observed variables remains fixed across domains. With additional assumptions, such as sufficient variability in the domain information, model identifiability can be achieved in these settings. Khemakhem et al. (2020); Hyvärinen & Morioka (2019; 2016)

However, such method is only limited to continuous variables. There have been several recent works that extend *ICA* to cases where the observed variables are discretized from latent continuous factors, such as in *binary ICA* Hyvärinen & Hoyer (2001). However, none of these provides any guarantees when both the latent and observed variables are discrete.

The inability to handle discrete latent variables poses significant challenges, especially in real-world applications. In practice, many scenarios feature only discrete variables, with latent variables considered to be discrete based on prior knowledge. For example, in disease diagnosis, physicians often make initial judgments about potential diseases based on a patient's symptoms. Here, "whether the patient has a disease" and "whether she exhibits symptoms" can be viewed as discrete latent and observed variables, respectively, both taking values of True/False. In Figure 1, the patient may have conditions like Gastritis, Asthma, or Anemia, represented as latent binary variables that are causally connected to symptoms, also modeled as binary variables $(X_1, X_2, X_3, X_4)$ in the figure. The goal is then to properly infer the unknown diseases from the observed symptoms. Other real-world examples of discrete-to-discrete model structures include **topic modeling** , where words in a document are generated from a distribution over discrete topics, with each document represented as a mixture of topics and each topic as a distribution over a finite vocabulary Blei et al. (2003); Goodman (1974); Pritchard et al. (2000); Collins & Lanza (2010). Another example is **survey data**, where responses to questions (e.g., agree/disagree levels) are discrete, and the underlying latent traits (such as personality or attitudes) are also modeled as discrete variables. This approach is commonly used when researchers believe the population consists of distinct, unobserved subgroups that explain the differences in survey responses, as seen in methods like *Latent Class Analysis (LCA)* Lazarsfeld & Henry (1968); Collins & Lanza (2010).

In this paper, we provide concrete identifiability guarantees for cases where both observed and latent variables are discrete. We derive identifiability results under various settings. First, we demonstrate that identifiability is readily achievable under the nonlinear ICA framework, which assumes an invertible mapping between the observed and latent variables. Next, we extend this framework to accommodate arbitrary mappings and establish sufficient conditions for identifiability in these more general settings. In each case, we show how incorporating multi-domain information can significantly relax the constraints on model parameters. Finally, we empirically validate our theoretical results using simulated data and demonstrate their practical applicability on multiple real-world datasets.

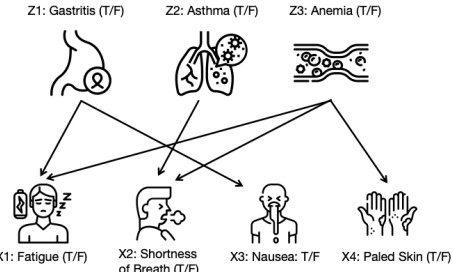

Figure 1: Modeling Binary Latent Diseases and Observed Symptoms

## 2 PROBLEM STATEMENT

Different from the continuous setting, where the goal is often to recover the values of latent variables and the functional relations between latent and observed variables—relying on strong structural assumptions such as the invertible mixing function and the smoothness of probability density functions (pdfs)—the goal in discrete settings typically focuses on identifying relations at the distributional level, emphasizing the recovery of relevant distributions through discrete probability mass functions (pmfs) rather than the specific values of the latent variables, which can be flexibly assigned. This distinction is well-established in models such as *LCA* and finite mixture models Lazarsfeld & Henry (1968); McLachlan & Peel (2000).

In this paper, the observed variables $\mathbf{X}$ are modeled as a mixture of conditional distributions based on the possible values of the latent variables $\mathbf{Z}$. For $K$ independent domains where the random variable $\mathbf{u}$ takes values $\mathbf{u}_k$, with $k \in \{1, \cdots, K\}$, we assume that the latent variables are dependent on $\mathbf{u}$. Similar to the continuous case, we assume that the marginal distributions $P(\mathbf{Z} \mid \mathbf{u})$ vary across domains, while the generating mechanism $P(\mathbf{X} \mid \mathbf{Z})$ remains fixed across all domains. Our primary goal is to identify both $P(\mathbf{X} \mid \mathbf{Z})$ and $P(\mathbf{Z} \mid \mathbf{u})$.

In line with *Latent Class Models (LCM)* Goodman (1974); McCutcheon (1987); Vermunt & Magidson (2002), we aim to establish *state-level identifiability* (Definition 1). Inspired by *nonlinear ICA*, we adapt the use of domain information to address the identifiability problem in a more general setting. Without loss of generality, we consider both the observed and latent variables to be binary, as categorical variables can be transformed into binary form using appropriate encoding techniques. The specific model settings and necessary assumptions are presented as follows.

MODEL SETTING

- Given $K$ independent domains, indexed by $\mathbf{u}_k, k \in \{1, \cdots, K\}$, we define $N$ binary observed variables, $\mathbf{X} = \{X_1, \cdots, X_N\}$, in each domain, with distribution denoted as: $P(\mathbf{X} = \mathbf{x} \mid \mathbf{u}), \mathbf{x} \in \{0, 1\}^N$.

- At the latent level, we introduce $D$ binary latent variables, $\mathbf{Z} = \{Z_1, \cdots, Z_D\}$, with a joint distribution given by: $P(\mathbf{Z} = \mathbf{z} \mid \mathbf{u}), \mathbf{z} \in \{0, 1\}^D$.

- The observed variables $\mathbf{X}$ are generated from the latent variables $\mathbf{Z}$ with $P(\mathbf{X} \mid \mathbf{u}) = \sum_{l=1}^{2^D} P(\mathbf{X} \mid \mathbf{Z} = \mathbf{z}_l) P(\mathbf{Z} = \mathbf{z}_l \mid \mathbf{u})$.

**Assumption 1.** *We assume that the variables satisfy the following conditions:*

1. *The effects of $\mathbf{Z}$ on each observed variable $X_i$ for $i \in \{1, \cdots, N\}$ are conditionally independent, such that $P(\mathbf{X} \mid \mathbf{Z}) = \prod_{i=1}^{N} P(X_i \mid \mathbf{Z})$.*

2. *For each $X_i \in \mathbf{X}$, the conditional distributions $P(X_i \mid \mathbf{Z})$ are the same across all domains, i.e., they are independent of the domain variable $\mathbf{u}$.*

3. *There exists a mapping $S : \mathbb{R}^N \to \mathbb{R}^m$ such that $\text{rank}(P(S(\mathbf{X}) \mid \mathbf{u})) = 2^D$ for any domain. For simplicity, in the following, we assume the subspace corresponds to the original variable space, with the number of variables still denoted as $N$.*

**Definition 1** (State-Level Identifiability). *Let $X$ be a random variable with a discrete set of possible states $\{s_1, s_2, \ldots, s_k\}$ and an associated probability distribution $P(X = s_i) = p_i$ for $i = 1, \ldots, k$, where $p_i \geq 0$ and $\sum_{i=1}^{k} p_i = 1$. The random variable $X$ is said to have **state-level identifiability** if its distribution is identifiable up to a permutation of its states. Formally, this means there exists a permutation $\sigma \in S_k$ such that:*

$$P_Y = \{p_{\sigma(1)}, p_{\sigma(2)}, \ldots, p_{\sigma(k)}\},$$

*where $\sigma$ is an element of the symmetric group $S_k$, representing all possible permutations of $\{1, 2, \ldots, k\}$.*

Based on Assumption 1, we can rewrite the distributions over $\mathbf{X}$ in each domain as the mixture of the conditional probabilities of individual $X_i \in \mathbf{X}$, as given by the following equation:

$$P(\mathbf{X} \mid \mathbf{u}) = \sum_{l=1}^{2^D} \prod_{i=1}^{N} P(X_i \mid \mathbf{Z} = \mathbf{z}_l) P(\mathbf{Z} = \mathbf{z}_l \mid \mathbf{u}) \tag{1}$$

In this paper we aim to provide identifiability conditions on the attributes as follows.

- The conditional distributions $P(X_i \mid \mathbf{Z}), i \in \{1, \cdots, N\}$.

- The conditional distributions $P(\mathbf{Z} \mid \mathbf{u})$.

- The marginal distributions $P(Z_j \mid \mathbf{u}), j \in \{1, \cdots, D\}$, when the latent variables are further assumed to be mutually independent.

## 3 IDENTIFIABILITY WITH ONE-TO-ONE MAPPING

With continuous variables, an invertible mapping from the latent space to the observed space is often assumed, where the change-of-variables rule can be used to recover distributional relationships between the latent and observed variables and establish identifiability. In this section, we demonstrate that under invertible mapping condition, identifiability established in the continuous case holds in the discrete case.

**Assumption 2.** *In each domain, for every value* $\mathbf{x}$ *of the observed variables* $\mathbf{X}$*, there exists a unique corresponding value* $\mathbf{z}$ *of the latent variables* $\mathbf{Z}$*, such that* $P(\mathbf{X} = \mathbf{x} \mid \mathbf{u}) = P(\mathbf{Z} = \mathbf{z} \mid \mathbf{u}) \neq 0$*, and vice versa.*

**Lemma 1.** *Under the invertibility assumption, the number of discrete variables* $D$ *must equal to the number of observed variables* $N$*.*

Under Assumption 2, Lemma 1 can be easily derived. Given this lemma, for simplicity, we replace all instances of $N$ with $D$ in this section. Since $\mathbf{X} = \mathbf{f}(\mathbf{Z})$, where $\mathbf{f} : \{0, 1\}^D \rightarrow \{0, 1\}^D$ is an invertible mapping, the distributions over the states of $\mathbf{X}$ and $\mathbf{Z}$ are also mapped one-to-one. We now present the identifiability results for $P(X_i \mid \mathbf{Z})$ and $P(\mathbf{Z} \mid \mathbf{u})$.

**Theorem 1.** *Given Assumptions 1 and 2, let* $d \in \{1, \ldots, 2^D\}$ *denote a state of* $\mathbf{X}$ *and* $\mathbf{Z}$*, such that* $P(\mathbf{X} = \mathbf{x}_d \mid \mathbf{u}) = P(\mathbf{Z} = \mathbf{z}_d \mid \mathbf{u})$*. Define* $x_{i,d}$ *as the value of* $X_i$ *in* $\mathbf{x}_d$*, i.e.,* $x_{i,d} \in \{0, 1\}$*. Then, for any* $i \in \{1, \ldots, D\}$ *and* $x_i \in \{0, 1\}$*, we have:*

$$P(X_i = x_i \mid \mathbf{Z} = \mathbf{z}_d) = \begin{cases} 1 & \text{if } x_i = x_{i,d}, \\ 0 & \text{otherwise}. \end{cases}$$

### 3.1 WITH FURTHER INDEPENDENCE CONDITION

In addition to the one-to-one mapping, it is common in representation learning to assume that the latent factors are mutually independent within each domain, which facilitates the recovery of the latent variables' properties Schölkopf et al. (2021); Hyvärinen & Morioka (2019); Ouyang & Xu (2022). In the discrete case, we demonstrate that by imposing similar assumptions as in 3, we can also further identify the marginal distributions of the latent variables.

**Assumption 3.** $P(\mathbf{Z} \mid \mathbf{u}) = \prod_{j=1}^{D} P(Z_j \mid \mathbf{u})$.

**Lemma 2.** *Under Assumptions 1, 2 and 3, we have*

$$P(\mathbf{X} \mid \mathbf{u}) = \prod_{i=1}^{N} P(X_i \mid \mathbf{Z}) \prod_{j=1}^{D} P(Z_j \mid \mathbf{u})$$

$$= \prod_{j=1}^{D} P(Z_j \mid \mathbf{u}). \tag{2}$$

*Then, for any* $j \in \{1, \cdots, D\}$*,* $P(Z_j \mid \mathbf{u})$ *is identifiable.*

*Proof.* Easily we can derive Theorem 2 from Equation 1 under the assumptions. After taking $\log$ on both sides of Equation 2, we have a set of linear equations for unknown parameters $P(Z_j \mid \mathbf{u}), j \in \{1, \cdots, D\}$. The linear equations have unique solutions if and only if $2^{D-1} \geq D$, which holds for all $D \geq 1$. We then prove the unique solution for the marginal distribution of $Z_j, \forall j \in \{1, \cdots, D\}$ in each domain. $\square$

In conclusion, when the mapping between $\mathbf{X}$ and $\mathbf{Z}$ is invertible, identifiability naturally holds for discrete variables without requiring additional constraints on the latent variables (or observed variables) or the number of domains. Specifically, we can straightforwardly establish the identifiability of $P(X_i \mid \mathbf{Z})$, $i \in \{1, \cdots, N\}$ and $P(\mathbf{Z} \mid \mathbf{u})$. Furthermore, if we impose the independence assumption on the latent variables, we can also identify the marginal probabilities $P(Z_j \mid \mathbf{u})$, $j \in \{1, \cdots, D\}$.

# 4 IDENTIFIABILITY WITH FLEXIBLE MAPPING

In this section, we aim to relax the constraint of having an invertible mapping between $\mathbf{X}$ and $\mathbf{Z}$. While such a constraint is beneficial for ensuring identifiability, as shown in the previous section, it is often impractical in real-world scenarios. In practice, the distributions of $\mathbf{X}$ and $\mathbf{Z}$ may not align, and their support sizes may differ significantly. To address this, we present identifiability results under the more flexible assumption that the mapping between observed and latent variables can be arbitrary. A formal definition of this flexible mapping is provided in Appendix A.2. We establish sufficient conditions for both local and strict identifiability and demonstrate how multi-domain information can help alleviate constraints on model parameters.

## 4.1 LOCAL IDENTIFIABILITY

**Definition 2.** *Rothenberg (1971)[Local Identifiability] A parametric model $\mathcal{M}(\theta)$, where $\theta \in \Theta$, is* **locally identifiable** *at $\theta_0 \in \Theta$ if:*

$$\exists \epsilon > 0 \text{ such that } \mathcal{M}(\theta) = \mathcal{M}(\theta_0) \implies \theta = \theta_0, \quad \forall \theta \in B_\epsilon(\theta_0) \cap \Theta,$$

*where $B_\epsilon(\theta_0) = \{\theta \in \Theta : \|\theta - \theta_0\| < \epsilon\}$ is the open ball of radius $\epsilon$ around $\theta_0$.*

We begin with sufficient conditions for local identifiability, which we define in Definition 2. Local identifiability has been widely studied in representation learning, especially in the context of *nonlinear ICA* Buchholz et al. (2022); Hyvarinen & Morioka (2017); Hyvärinen & Morioka (2019).

Local identifiability is essential for many real-world problems, where exploring the entire parameter space is often impractical or unnecessary. For instance, in fMRI data analysis, where *nonlinear ICA* is often employed to disentangle brain signals, the focus is typically on identifying activity in specific brain regions rather than modeling the entire brain's activity. In such cases, the ability to make accurate predictions around the estimated parameters is more important than achieving strict identifiability across the entire model.

For clarity, we introduce additional notations of free parameters based on Equation 1. Let $\alpha_{l,k} = P(\mathbf{Z} = \mathbf{z}_l \mid \mathbf{u} = \mathbf{u}_k)$, where $l \in \{1, \ldots, 2^D - 1\}$, represent the free parameters used to characterize the distributions of the latent variables, and let $\beta_{i,l} = P(X_i = 0 \mid \mathbf{Z} = \mathbf{z}_l)$, where $l \in \{1, \ldots, 2^D\}$ represent the conditional probabilities of the observed variables. We then compute the partial derivatives of $P(\mathbf{X} \mid \mathbf{u})$ with respect to these free parameters, which leads to the construction of the following Jacobian matrix. $\mathbf{J}$ here has $(2^N - 1) \cdot K$ rows and $2^D \cdot N + (2^D - 1) \cdot K$ columns, i.e.,

$$\mathbf{J} = (\mathbf{J}_{\alpha_{(1,1)}}, \mathbf{J}_{\alpha_{(2,1)}}, \cdots, \mathbf{J}_{\alpha_{(2^D-1,1)}}, \cdots, \mathbf{J}_{\alpha_{(2^D-1,K)}}, \mathbf{J}_{\beta_{(1,1)}}, \cdots, \mathbf{J}_{\beta_{(N,2^D)}})$$

$$where \ \mathbf{J}_{\alpha_{l,k}} = \frac{\partial P(\mathbf{X} \mid \mathbf{u})}{\partial \alpha_{l,k}} = \prod_{i=1}^{N} P(X_i \mid \mathbf{Z} = \mathbf{z}_l) - \prod_{i=1}^{N} P(X_i \mid \mathbf{Z} = \mathbf{z}_{2^D}) \tag{3}$$

$$\mathbf{J}_{\beta_{i,l}} = \frac{\partial P(\mathbf{X} \mid \mathbf{u})}{\partial \beta_{i,l}} = (-1)^{X_i} \prod_{p=1, p\neq i}^{N} P(X_p \mid \mathbf{Z} = \mathbf{z}_l) P(\mathbf{Z} = \mathbf{z}_l \mid \mathbf{u}).$$

**Assumption 4.** *We make the following assumptions:*

1. $2^N K \geq 2^D N + 2^D K$.

2. *The free parameters $\{\alpha_{(1,1)}, \cdots, \alpha_{(2^D-1,K)}, \beta_{(1,1)}, \cdots, \beta_{(N,2^D)}\}$ are all positive.*

We then have the following theorem:

**Theorem 2.** *Given Assumptions 1 and 4, for any $\{i, j\}$, $P(X_i \mid \mathbf{Z})$ and $P(Z_j \mid \mathbf{u})$ are locally identifiable if and only if the corresponding Jacobian matrix $\mathbf{J}$ in Equation 3 has full column rank.*

The local identifiability result relies on the nonsingularity of the Fisher information matrix, which characterizes the curvature of the likelihood function of $P(\mathbf{X} \mid \mathbf{u})$. This nonsingularity is ensured by the full column rank condition of the Jacobian $\mathbf{J}$ defined here. Moreover, we observe that multi-domain information plays a critical role in Assumption 4, serving as an essential component of the sufficient conditions for ensuring identifiability. Specifically, with sufficient domain information, identifiability can be guaranteed as long as the number of observed variables exceeds the number of latent variables.

### 4.1.1 WITH FURTHER INDEPENDENCE CONDITION

If we further assume that the latent variables are conditionally independent in each domain, we can rewrite the relation between $P(\mathbf{X} \mid \mathbf{u})$ and $P(\mathbf{Z} \mid \mathbf{u})$ in Equation 4 as:

$$P(\mathbf{X} \mid \mathbf{u}) = \sum_{l=1}^{2^D} \prod_{i=1}^{N} P(X_i \mid \mathbf{Z} = \mathbf{z}_l) \prod_{j=1}^{D} P(Z_j = z_{lj} \mid \mathbf{u}). \tag{4}$$

Here $\mathbf{z}_l$ is a $D$-dimensional vector, and $z_{lj} \in \mathbf{z}_l$. Like above, we denote the free parameters in the equations as $\gamma_{j,k} = P(Z_j = 0 \mid \mathbf{u} = \mathbf{u}_k)$, and $\beta_{i,l} = P(X_i = 0 \mid \mathbf{Z} = \mathbf{z}_l)$, where $j \in \{1, \cdots, D\}$, $i \in \{1, \cdots, N\}$, $k \in \{1, \cdots, K\}$, and $l \in \{1, \cdots, 2^D\}$.

We then define the associated Jacobian $\mathbf{J}^{Ind}$ as

$$\mathbf{J}^{Ind} = (\mathbf{J}_{\gamma_{(1,1)}}, \mathbf{J}_{\gamma_{(2,1)}}, \cdots, \mathbf{J}_{\gamma_{(D,1)}}, \cdots, \mathbf{J}_{\gamma_{(D,K)}}, \mathbf{J}_{\beta_{(1,1)}}, \cdots, \mathbf{J}_{\beta_{(N,2^D)}}),$$

$$where \ \ \mathbf{J}_{\gamma_{j,k}} = \frac{\partial P(\mathbf{X} \mid \mathbf{u})}{\partial \gamma_{j,k}} = \sum_{l=1}^{2^D} (-1)^{z_{lj}} \prod_{i=1}^{N} P(X_i \mid \mathbf{Z} = \mathbf{z}_l) \prod_{q=1, q \neq j}^{D} P(Z_q = z_{lq} \mid \mathbf{u}), \tag{5}$$

$$\mathbf{J}_{\beta_{i,l}} = \frac{\partial P(\mathbf{X} \mid \mathbf{u})}{\partial \beta_{i,l}} = (-1)^{X_i} \prod_{p=1, p \neq i}^{N} P(X_p \mid \mathbf{Z} = \mathbf{z}_l) \prod_{j=1}^{D} P(Z_l = z_{lj} \mid \mathbf{u}).$$

The Jacobian $\mathbf{J}^{Ind}$ here has $(2^N - 1) \cdot K$ rows and $2^D \cdot N + D \cdot K$ columns.

Under the following assumptions, we derive the necessary and sufficient conditions for local identifiability, assuming independence of the latent variables in Theorem 3.

**Assumption 5.** *We make the following assumptions:*

1.  $K(2^N - 1) \geq 2^D \cdot N + KD$

2.  *The free parameters $\left\{\gamma_{(1,1)}, \cdots, \gamma_{(D,K)}, \beta_{(1,1)}, \cdots, \beta_{(N,2^D)}\right\}$ are all positive.*

**Theorem 3.** *Under the Assumptions 1, 3 and 5, for any $\{i, j\}$, $P(X_i \mid \mathbf{Z})$ and $P(Z_j \mid \mathbf{u})$ are locally identifiable if and only if the corresponding Jacobian matrix $\mathbf{J}^{Ind}$ has full column rank*

The underlying idea behind this theorem is similar to that of Theorem 2.

In this section, we have established the necessary and sufficient conditions for local identifiability, ensuring that the model exhibits distinct local behavior around the ground truth. Additionally, we demonstrated how multi-domain information can relax these conditions, as shown in Assumption 4 and 5, compared to the scenario with only a single domain (K=1).

### 4.2 STRICT IDENTIFIABILITY

Although local identifiability is often sufficient in many cases, complex applications may require guarantees of strict identifiability, where the model's parameters can be uniquely determined by the observed distributions $P(\mathbf{X} \mid \mathbf{u})$ across domains.

Ensuring strict identifiability is particularly challenging in the discrete case with flexible mappings, where we can not simply align observed and latent variable distributions and recover their functional

relations, as is often done in nonlinear ICA models with continuous variables. However, regardless of the mapping structure between observed and latent variables, the relation between observed distributions and the products of conditional distributions $P(\mathbf{X} \mid \mathbf{Z})$ and marginal distributions $P(\mathbf{Z} \mid \mathbf{u})$ always holds in every domain as shown in Equation 1. Fortunately, with assumptions on the independent generating processes of $\mathbf{X}$ in Assumption 1, the problem can be simplified and reformulated into an N-way array decomposition problem.

In this section, we demonstrate that under this reformulation, strict identifiability of model parameters can be established in a highly general setting, without requiring specific structure or distributional assumptions. Furthermore, we empirically show that when latent variables are independent, the required number of observed variables and domains can be significantly reduced.

We begin by illustrating how this problem can be reformulated as a multilinear decomposition of N-way arrays as shown in Lemma 3.

**Definition 3** (Conditional probability matrix)**.** *For any $X_i \in \mathbf{X}$, its corresponding conditional distribution given the latent variables $\mathbf{Z}$ can be represented as a $2 \times 2^D$ matrix $A^i$. The l-th column of this matrix contains the vector of conditional probabilities $P(X_i = 0 \mid \mathbf{Z} = \mathbf{z}_l)$ and $P(X_i = 1 \mid \mathbf{Z} = \mathbf{z}_l)$, where $l \in \{1, \ldots, 2^D\}$.*

**Definition 4** (Joint probability matrix)**.** *Similarly, we can form the joint probabilities of $\mathbf{Z}$ in different domains as a $K \times 2^D$ matrix $B$, with k-th row denoting the joint probabilities over the $2^D$ states of $\mathbf{Z}$ in domain $k$.*

**Lemma 3.** *Based on the Definitions 3 and 4, Equation 1 can be rewritten as:*

$$P(\mathbf{X} \mid \mathbf{u}) = \sum_{l=1}^{2^D} (\otimes_{i=1}^N A^i_{(\cdot,l)}) \tilde{\otimes} B_{(\cdot,l)}, \tag{6}$$

*where $\otimes$ denotes the outer product of vectors and $\tilde{\otimes}$ denotes the outer product between a tensor and a vector, and $A^i_{(\cdot,l)}, B_{(\cdot,l)}$ denote the l-th column of matrices $A^i, B$ . The term $P(\mathbf{X} \mid \mathbf{u})$ here is reshaped as a tensor of size $(\underbrace{2, 2, \ldots, 2}_{2^N \text{ times}}, K)$.*

**Theorem 4.** *Under Assumption 1, if $N \geq 2^D, K \geq 2^{D+1} - N$, for any $i \in \{1, \cdots, N\}$ and $k \in \{1, \cdots, K\}$, $P(X_i \mid \mathbf{Z})$ and $P(\mathbf{Z} \mid \mathbf{u})$ are identifiable.*

Similarly, if we further assume that the latent factors are mutually independent within each domain, we can also recover the marginal distribution of each latent variable, provided that the distributions of the latent variables $Z_i$ and $Z_j$ are distinct in every domain..

**Assumption 6.** *For $\forall i, j, k, P(Z_i = 0 \mid \mathbf{u}) \neq P(Z_j = 0 \mid \mathbf{u})$*

**Theorem 5.** *Under Assumptions 1, 3, and 6, if $N \geq 2^D, K \geq 2^{D+1} - N$, for any $j \in \{1, \cdots, D\}$, $P(Z_j \mid \mathbf{u})$ is also identifiable.*

In conclusion, the model is strictly identifiable under simple conditions on the model parameters, regardless of the specific mapping structures between the observed and latent variables. Multi-domain information plays a crucial role in relaxing these constraints, making identifiability more achievable in practical settings. Specifically, when there is only a single domain, the condition $N \geq 2^{D+1} - 1$ must be satisfied to ensure identifiability. However, with sufficient multi-domain information, this requirement is significantly relaxed to $N \geq 2^D$, demonstrating how the availability of multiple domains can reduce the burden of parameter constraints and enhance the applicability of the model.

Moreover, when the latent variables are independent, we demonstrate empirically in the following section that the number of required domains $K$ outlined in Theorem 5 can be further greatly relaxed.

## 5 EXPERIMENTS

In this section we empirically demonstrate the validity of our identifiability results by testing them on both simulated and real-world datasets.

## 5.1 SIMULATION RESULTS

We first validate our identifiability results under flexible mappings using simulated binary datasets with multiple domains. For clarity and illustration purposes, we present the cases where the latent variables are assumed to be independent, enabling us to also test the accuracy of their marginal distributions.

**Data** We begin by simulating the free parameters for the data. For each variable, we ensure that the sum of probabilities across each domain equals 1. Specifically, given the number of observed variables $N$, the number of domains $K$, and the number of latent variables $D$, we first generate the matrix of free parameters for the marginal distributions $P(Z_j \mid \mathbf{u})$, where $Z_j \in \mathbf{Z}$, with a total size of $(D, K)$. Additionally, we generate the matrix of free parameters for the conditional distributions $P(X_i \mid \mathbf{Z})$, where $X_i \in \mathbf{X}$, with a total size of $(N, 2^D)$. Using these parameters, we then compute the distributions of the observed variables in each domain, according to $P(\mathbf{Z} \mid \mathbf{u})$ and $P(\mathbf{X} \mid \mathbf{Z})$, following the model described above.

**Evaluation** In our experiments, we use the Kullback-Leibler (KL) divergence Kullback & Leibler (1951) to measure the difference between the estimated and true distributions. For the latent variables in domain $k$, the KL divergence is defined as $KL_{\mathbf{Z}|\mathbf{u}} = \frac{1}{D} \sum_{j=1}^{D} \left( P_{jk}^{\text{es}} \log \left( \frac{P_{jk}^{\text{es}}}{P_{jk}^{\text{tr}}} \right) + (1 - P_{jk}^{\text{es}}) \log \left( \frac{1 - P_{jk}^{\text{es}}}{1 - P_{jk}^{\text{tr}}} \right) \right)$, where $P_{jk}^{\text{es}}$ and $P_{jk}^{\text{tr}}$ denote the estimated and true probabilities $P(Z_j = 0 \mid \mathbf{u})$. For the probabilities of observed variables given latent configuration $\mathbf{Z}$, the KL divergence is calculated as $KL_{\mathbf{X}|\mathbf{Z}} = \frac{1}{2^D N} \sum_{l=1}^{2^D} \sum_{i=1}^{N} \left( P_{il}^{\text{es}} \log \left( \frac{P_{il}^{\text{es}}}{P_{il}^{\text{tr}}} \right) + (1 - P_{il}^{\text{es}}) \log \left( \frac{1 - P_{il}^{\text{es}}}{1 - P_{il}^{\text{tr}}} \right) \right)$, where $P_{il}^{\text{es}}$ and $P_{il}^{\text{tr}}$ represent the estimated and true probabilities $P(X_i = 0 \mid \mathbf{Z} = \mathbf{z}_l)$.

### 5.1.1 LOCAL IDENTIFIABILITY

We first demonstrate local identifiability results using simulated datasets. We test various scenarios with different number of latent variables $D$, observed variables $N$ and domains $K$. Starting from the ground truth, we introduce small perturbations and run 100 experiments per case with different initializations to avoid numerical errors. We then collect the results with the highest likelihood and compute the KL divergence between these and the ground truth.

**Results** Here we present the results for $KL_{\mathbf{X}|\mathbf{Z}}$ and $KL_{\mathbf{Z}|\mathbf{u}}$ for the case where $D$ is 3, $N$ ranging from 3 to 5, and $K$ ranging from 1 to 6. This showcases the empirical minimum number of observed variables required as the number of domains changes. For improved visualization, the KL divergence values presented here are scaled by a factor of $e^3$. Entries with values less than $0.1$ are highlighted in both tables, indicating cases where empirical identifiability holds. In comparison with the theoretical results, in Table 1 and 2, entries corresponding to the minimum required number of theoretically required domains in Theorem 3 for a given $N$ are underlined. We can see that the theoretical results closely align with the empirical evidence where both $P(X_i \mid \mathbf{Z})$ and $P(Z_j \mid \mathbf{u})$, with $i \in \{1, \cdots, N\}, j \in \{1, \cdots, D\}$, are locally identifiable.

| $K/N$ | 3 | 4 | 5 |
|---|---|---|---|
| 1 | 0.1818 | 0.1852 | 8.849e-02 |
| 2 | 0.3042 | 6.936e-02 | 4.863e-03 |
| 3 | 0.1490 | 5.504e-03 | 1.425e-06 |
| 4 | 0.1256 | 4.509e-06 | 3.697e-07 |
| 5 | 7.078e-02 | 2.047e-06 | 9.625e-08 |
| 6 | 8.058e-02 | 3.034e-07 | 1.920e-08 |

Table 1: Local Identifiability: $KL_{\mathbf{X}|\mathbf{Z}}$ with $D = 3$

| $K/N$ | 3 | 4 | 5 |
|---|---|---|---|
| 1 | 0.2711 | 8.778e-02 | 0.4623 |
| 2 | 0.2805 | 0.1578 | 1.148e-02 |
| 3 | 7.662e-02 | 3.168e-03 | 1.425e-06 |
| 4 | 0.1997 | 1.812e-05 | 4.670e-07 |
| 5 | 6.023e-02 | 1.235e-05 | 1.562e-07 |
| 6 | 9.576e-02 | 3.429e-07 | 9.969e-08 |

Table 2: Local Identifiability: $KL_{\mathbf{Z}|\mathbf{u}}$ with $D = 3$

| $K$/$N$ | | 4 | | 5 | | 6 | | 7 | | 8 |
|---|---|---|---|---|---|---|---|---|---|---|
| | 4 | 0.4744 | 2 | 0.6101 | 2 | 0.1122 | 1 | 0.6550 | 1 | 0.2274 |
| - | 5 | 4.001e-03 | 3 | 1.217e-02 | 3 | 4.503e-04 | 2 | 1.191e-04 | 2 | 1.153e-04 |

Table 3: Strict Identifiability: $KL_{\mathbf{Z}|\mathbf{u}}$ with $D = 3$

| $K$/$N$ | | 4 | | 5 | | 6 | | 7 | | 8 |
|---|---|---|---|---|---|---|---|---|---|---|
| | 4 | 0.7752 | 2 | 0.4711 | 2 | 0.2379 | 1 | 1.124 | 1 | 0.2024 |
| - | 5 | 4.404e-04 | 3 | 1.349e-02 | 3 | 1.783e-04 | 2 | 2.325e-05 | 2 | 1.627e-04 |

Table 4: Strict Identifiability: $KL_{\mathbf{X}|\mathbf{Z}}$ with $D = 3$

### 5.1.2 STRICT IDENTIFIABILITY

To verify our results on strict identifiability, we randomly initialize the starting points and calculate the KL divergence between the final estimated distributions and the true distributions. For each unique combination of $\{D, N, K\}$, we reinitialize the starting points 100 times and select the one that maximizes the likelihood. The KL divergence $KL_{\mathbf{X}|\mathbf{Z}}$ and $KL_{\mathbf{Z}|\mathbf{u}}$, when $D = 3$, are summarized in Tables 3 and 4. For simplicity, we present the minimum number of required domains $K$ for the distributions to be identifiable for each $N$, as shaded in the Tables. When $N = 2$ or $N = 3$, more than 20 domains were required for parameter identification, so we consider these cases unidentifiable and omit them from the results.

**Results** The simulation results validate the sufficiency of the domain requirements outlined in Theorem 4. Moreover, under the assumption of mutual independence among latent variables within each domain, we empirically show that the required number of domains is significantly relaxed compared to the conditions specified in Theorem 5. For example, with 3 latent variables and 7 or 8 observed variables, only 2 domains are necessary to identify the unobserved distributions as shown in Table 3 and 4. These findings strongly support the benefit of leveraging multiple domains, as they can substantially reduce the overall data requirements for strict identifiability.

## 5.2 REAL WORLD DATA

### 5.2.1 DATA

We apply our method to two real-world datasets to demonstrate its applicability, assuming that the latent variables are independent in a given domain. The first dataset, **Big Five**, is a psychological dataset collected through an online personality test Howard & Howard (2010). The dataset consists of 50 discrete variables and approximately 20,000 data points. Each of the Big Five personality dimensions—Openness, Conscientiousness, Extraversion, Agreeableness, and Neuroticism (O-C-E-A-N)—is measured using 10 indicators. For illustration, we select 2 observed variables from each dimension and divide them into 2 categories, resulting in a total of 10 observed binary variables. We test several potential values of $D$, ultimately selecting 5 latent variables with the highest likelihood. Using age and gender as domain-defining variables, we divide the dataset into 9 domains.

The second dataset, **NASDAQ-listed stocks**, contains year-to-date (YTD) gain information for 8 different NASDAQ stocks from 2003 to 2023 Mooney. Although the data are inherently continuous, we binarize the time series based on median values over two-year intervals. We treat each interval as a separate domain, resulting in 11 domains. We find out that the model achieves the highest likelihood with 6 latent variables.

### 5.2.2 EVALUATION

After obtaining the estimated distributions, we evaluate the effect of $Z_j$ on $X_i$ for any pair of latent and observed variables $(Z_j, X_i)$. Specifically, we test the significance of the difference between $P(X_i = 0 \mid Z_j = 0, \tilde{\mathbf{Z}})$ and $P(X_i = 0 \mid Z_j = 1, \tilde{\mathbf{Z}})$, averaging over the possible values of $\tilde{\mathbf{Z}}$, which represents the other latent variables. This approach allows us to isolate the effect of $Z_j$

on $X_i$, ensuring that the observed differences on $X_i$'s conditional distribution is due to $Z_j$ while accounting for the influence of other latent factors.

Figures 2 and 3 illustrate this relationship, showing edges between $Z_j$ and $X_i$ when the difference is statistically significant.

### 5.2.3 RESULTS

For the **Big Five** dataset, shown in Figure 2, we observe that the observed variables measuring similar personality traits are often influenced by the same latent variables. For instance, $N_1$ and $N_2$, which assess Neuroticism, are both significantly influenced by $Z_3$. Similarly, $A_1$ and $A_2$, which measure Agreeableness, are primarily influenced by $Z_1$. Notably, since the observed variables for each trait are selected to reflect opposite aspects of the trait—one positive and the other negative—this is mirrored in the estimated effects from their shared latent variable. Specifically, one variable tends to receive a positive influence, while the other receives a negative influence from the same latent factor. This pattern is consistent with the findings reported in Dong et al. (2023).

For the **NASDAQ-listed stocks** dataset, shown in Figure 3, we similarly observe that stocks within the same sub-sector tend to be influenced by the same latent variables. For example, *AMZN*, *INTC*, and *MSFT*, all technology stocks, are significantly influenced by both $Z_2$ and $Z_3$. To further support this observation, we calculated the correlation between $P(Z_2 = 0)$ and the returns of the NASDAQ-100 technology sector over the years, based on the methodology from Nasdaq, which resulted in a p-value of 0.0355. This reveals a strong relationship, indicating that over the past 20 years, the distributions of $Z_2$ reflect the growth and performance of the technology sector.

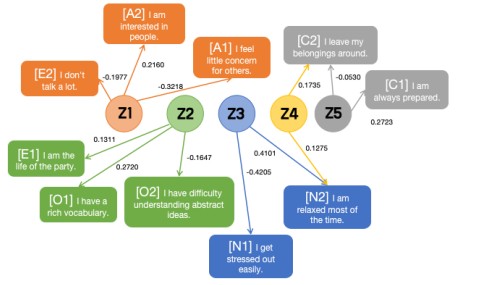
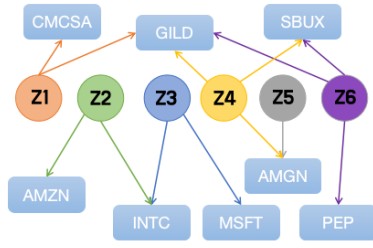

Figure 2: Big Five                    Figure 3: NASDAQ Stocks

## 6 DISCUSSION

To provide a clearer picture, we summarize the conditions for our main identifiability results as follows.

1. One-to-one mapping + Assumption 1,2: strict identifiability on $P(X_i \mid \mathbf{Z})$ and $P(\mathbf{Z} \mid \mathbf{u})$
2. One-to-one mapping + Assumption 1,2,3: strict identifiability on $P(X_i \mid \mathbf{Z})$ and $P(Z_j \mid \mathbf{u})$
3. Flexible mapping + Assumption 1,4: local identifiability on $P(X_i \mid \mathbf{Z})$ and $P(\mathbf{Z} \mid \mathbf{u})$
4. Flexible mapping + Assumption 1,3,5: local identifiability on $P(X_i \mid \mathbf{Z})$ and $P(Z_j \mid \mathbf{u})$
5. Flexible mapping + Assumption 1: strict identifiability on $P(X_i \mid \mathbf{Z})$ and $P(\mathbf{Z} \mid \mathbf{u})$
6. Flexible mapping + Assumption 1,3,6: strict identifiability on $P(X_i \mid \mathbf{Z})$ and $P(Z_j \mid \mathbf{u})$

One potential worry is that whether domain information is widely obtainable in real-world scenarios. In practice, however, domain distinctions can be often derived from various sources, such as labels across populations, experimental conditions, time periods, geographic regions, or demographic groups (e.g., age, gender, or socioeconomic status). These distinctions are integral to many datasets, making the theorem highly adaptable to a wide range of practical applications.

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

# A  APPENDIX

## A.1  BACKGROUND KNOWLEDGE

Here we provide additional information on *nonlinear ICA* and latent class models, which are closely related to the topic in this paper.

### A.1.1  ICA

*ICA* has been widely explored in the past decades, due to its ability in recovering the latent distributions and their structure connected to observed variables. *ICA* under linearity assumptions was first studies, which assumes that the latent variables $\{s_1, ..., s_d\}$ are transformed through an unknown mixing matrix $A$ to the observed data $\{x_1, ..., x_n\} = A\mathbf{s}$. *Linear ICA* is able to recover the mixing matrix $A$ and the $\mathbf{s}$ up to permutation and scaling, with the assumptions that $\mathbf{s}$ are independent and there is no more than one source following Gaussian distributions.

*Nonlinear ICA* generalizes the above results by considering the invertible mapping $\mathbf{x} = f(\mathbf{s})$. However, it is shown that there is always a transformation such that $\mathbf{z} = g(\mathbf{x})$ has a uniform distribution with independent $z_i \in \mathbf{z}$. It seems that the latent factors are not identifiable given the nonlinear structure. However, it is then discovered that identifiability can be established if there is a nonstationary temporal structure of the observed data, or the independent components. It is then generalized into the identifiability given just auxiliary variables that can modulates the latent components. In their setting, the components are dependent on the auxiliary variable but independent of each other, where $p(\mathbf{s} \mid \mathbf{u}) = \prod_i p_i(s_i \mid \mathbf{u})$. The auxiliary variable here can be widely defined, such as the index, class label, or domain knowledge. The theory shows that the $f$ and $\mathbf{s}$ are identifiable up to component-wise invertible transformations under some regularity constraints. Domain information is then widely used in various *ICA* identification tasks, with which we can separate the static parts from the changing parts and recover them. However, there are very few *ICA* research done on discrete variables. One reason is that, in the discrete case, some classical assumptions underlying *ICA* can be violated. For example, most *ICA* assumes smooth probability dense functions and are second-order differentiable. This obviously does not hold in the discrete case. Also, most of the *ICA* methods aim at recovering the values of latent variables, but for discrete variables we are only interested in recovering the distributions since the values of discrete variables do not make much sense. In conclusion, for discrete variables we must establish different principles for identifiability.

### A.1.2 LATENT CLASS MODELS

Latent class (LC) modeling was initially introduced by Lazarsfeld and Henry in 1968, especially designed for formulating latent attitudinal variables from dichotomous survey items. Different from factor analysis which tries to tackle mixture problems on continuous latent variables, LC models assume that the latent variables are categorical. Our argument in this paper borrows ideas from the recent developments in the identifiability of cognitive diagnosis models (CDMs). In CDM, each latent category corresponds to a distinct vector $\mathbf{z} = (z_1, \cdots, z_D) \in \mathcal{D} = \{0,1\}^D$, where $z_1, \cdots, z_D$ are all binary. The vector $\mathbf{z}$ denotes a unique latent profile with $z_j = 1$ implying the mastery of the subject on the $k-$th latent attribute and $z_j = 0$ implying the deficiency of it. The number of latent classes is denoted as $C = 2^D$ and the latent class membership as $c \in \{0, \cdots, 2^D - 1\}$. The response is denoted as $\mathbf{X} = (X_1, \cdots, X_N)$, and the latent class membership probabilities summarized as $\eta = P(\mathbf{Z} = c)$, the conditional probabilities $\theta_{jx_jc} = P(X_j = x_j \mid c)$ for getting response value $x_j$ in item $j$. Without loss of generosity, in this paper we treat the variables as binary (ref here). It also assumes that conditional on the latent class variables, the manifest variables are mutually independent, i.e. $P(X_i, X_j \mid \mathbf{Z}) = P(X_i \mid \mathbf{Z})P(X_j \mid \mathbf{Z}), \forall i \neq j$. Then the probability mass function for $\mathbf{X}$ can be written as $P(\mathbf{X} = \mathbf{x} \mid \eta, \mathbf{\Theta}) = \sum_{c=0}^{2^D-1} \eta_c \prod_{j=1}^{N} \theta_{jx_jc}$. Both the local and strict identifiability has then been proved regarding this model. For local identifiability, it sets the constraint that $2^N \geq 2^D(N+1)$ as the necessary condition. For strict identifiability, it furthermore imposes strong structure assumptions between the observed and latent variables, where each observed variable must have at least two pure children for the recovery of the distribution of latent variables.

### A.2 FLEXIBLE MAPPING

Here we give the definition of flexible mapping based on Hyvarinen & Morioka (2017); Khemakhem et al. (2020).

**Definition 5.** *A flexible mapping refers to a nonlinear function* $\mathbf{f}$*, which describes the relationship between a set of latent variables* $\mathbf{Z}$ *and observed variables* $\mathbf{X}$*, without the stringent assumptions of linearity, invertibility, or smoothness.*

*Mathematically, a flexible mapping can be written as:*

$$\mathbf{X} = \mathbf{f}(\mathbf{Z}; \theta)$$

*where:*

- $\mathbf{X} \in \mathbb{R}^N$ *is the vector of observed variables.*

- $\mathbf{Z} \in \mathbb{R}^D$ *is the vector of latent variables.*

- **f** *is a potentially highly nonlinear function parameterized by θ, which could represent the parameters of a neural network or another flexible functional form.*

## A.3 PROOFS

### A.3.1 PROOF OF THEOREM 1

*Proof.* Given $P(\mathbf{X} = \mathbf{x}_d \mid \mathbf{u}) = P(\mathbf{Z} = \mathbf{z}_d \mid \mathbf{u}) \neq 0$ for $d \in \{1, \cdots, 2^D\}$, and $\mathbf{x}_d = \{x_1^d, \cdots, x_N^d\}$, we have

$$P(\mathbf{X} = \mathbf{x}_d \mid \mathbf{u}) = \prod_{i=1}^{N} P(X_i = x_i \mid \mathbf{Z}) P(\mathbf{Z} = \mathbf{z}_d \mid \mathbf{u})$$

$$\Leftrightarrow \prod_{i=1}^{N} P(X_i = x_i^d \mid \mathbf{Z} = \mathbf{z}_d) = 1$$

(7)

$P(X_i = x_i^d \mid \mathbf{Z} = \mathbf{z}_d) = 1$ follows easily, the other possible values $X_i$ may take therefore has probability 0. We can then get the result as shown in Theorem 1. □

### A.3.2 PROOF OF THEOREM 2

*Proof.* Following the proof in Ouyang & Xu (2022), we define the likelihood function as

$$l(\mathbf{X}; \boldsymbol{\alpha}, \boldsymbol{\beta}, k) = \prod_{\mathbf{x} \in \Omega} P(\mathbf{X} = \mathbf{x} \mid \mathbf{u})$$

where $\Omega = \{0, 1\}^N$, $\boldsymbol{\alpha} = \{\alpha_{lk}\}_{l=1,k=1}^{l=2^D-1,k=K}$, and $\boldsymbol{\beta} = \{\beta_{il}\}_{i=1,l=1}^{i=N,l=2^D}$. We then denote the set of all parameters as

$$\boldsymbol{\eta} = \{\boldsymbol{\alpha}, \boldsymbol{\beta}\}.$$

The corresponding Fisher information matrix can be written as

$$\mathbb{E}\left[\left(\frac{\partial \log l}{\partial \boldsymbol{\eta}}\right)\left(\frac{\partial \log l}{\partial \boldsymbol{\eta}}\right)^T\right]$$

$$= \mathbb{E}\left[\left(\sum_{\mathbf{x} \in \Omega} \frac{\mathbb{I}\{\mathbf{X} = \mathbf{x}\} \frac{\partial P(\mathbf{X} = \mathbf{x})}{\partial \boldsymbol{\eta}}}{P(\mathbf{X} = \mathbf{x})}\right)\left(\sum_{\mathbf{x} \in \Omega} \frac{\mathbb{I}\{\mathbf{X} = \mathbf{x}\} \frac{\partial P(\mathbf{X} = \mathbf{x})}{\partial \boldsymbol{\eta}}}{P(\mathbf{X} = \mathbf{x})}\right)^T\right]$$

$$= \sum_{\mathbf{x} \in \Omega} \frac{1}{P(\mathbf{X} = \mathbf{x})}\left(\frac{\partial P(\mathbf{X} = \mathbf{x})}{\partial \boldsymbol{\eta}}\right)\left(\frac{\partial P(\mathbf{X} = \mathbf{x})}{\partial \boldsymbol{\eta}}\right)^T$$

$$= \mathbf{J}^T \begin{pmatrix} \frac{1}{P(\mathbf{X} = \mathbf{x}_1)} & 0 & \cdots & 0 \\ 0 & \frac{1}{P(\mathbf{X} = \mathbf{x}_2)} & \cdots & 0 \\ \vdots & \vdots & \ddots & \vdots \\ 0 & 0 & \cdots & \frac{1}{P(\mathbf{X} = \mathbf{x}_s)} \end{pmatrix} \mathbf{J}.$$

According to Theorem 1 of Rothenberg (1971), the free parameters in a model are locally identifiable if and only if the Fisher information matrix is non-singular. Given that the Fisher information matrix is non-singular if and only if **J** has full column rank as the equation shows, we learn that the free parameters $\boldsymbol{\alpha}$ and $\boldsymbol{\beta}$ are locally identifiable if and only if **J** has full column rank, with the number of equations (rows) more than the number of free parameters (columns). □

### A.3.3 PROOF OF THEOREM 3

Similar to the above proof, except for the free parameters in this case are $\boldsymbol{\lambda} = \{\boldsymbol{\gamma}, \boldsymbol{\beta}\}$. We then get the corresponding conditions for local identifiability,

### A.3.4 PROOF OF THEOREM 4

With $P(\mathbf{X} \mid \mathbf{u})$ expressed in Lemma 3, the original problem is then transformed into the multilinear decomposition problem. According to Theorem 3 in Sidiropoulos & Bro (2000), $A^1, \cdots, A^N, B$ can be uniquely solved up to permutation and scaling of columns, if the conditions in 4 holds. Provided that the sum of each column in $A^1, \cdots, A^N$ is 1 in our case, the scales for the matrices are fixed. Then the state-level identifiability of $P(X_i \mid \mathbf{Z}), i \in \{1, \cdots, N\}$ and $P(\mathbf{Z} \mid \mathbf{u})$ are ensured.

### A.3.5 PROOF OF THEOREM 5

*Proof.* Under Assumption 6, we arrange the probabilities $P(\mathbf{Z} = \mathbf{z} \mid \mathbf{u})$ in ascending order, where $\mathbf{z} \in \{0,1\}^D$ represents the possible states of the latent variables. Assume without loss of generality that $P(Z_j = 0 \mid \mathbf{u}) \leq P(Z_j = 1 \mid \mathbf{u})$. We relabel the latent variables $Z_1, Z_2, \ldots, Z_D$ such that $Z_i$ represents the latent variable with the $i$-th smallest value of $P(Z_i = 0 \mid \mathbf{u})$ across all variables.

With this ordering, we can express the ratio of the minimum to the maximum probability over the latent space as:

$$\frac{\min P(\mathbf{Z})}{\max P(\mathbf{Z})} = \frac{P(Z_1 = 0 \mid \mathbf{u})}{P(Z_D = 0 \mid \mathbf{u})}.$$

By exploiting this ordered structure and the fact that the latent variables are independent, we can systematically identify the marginal distributions $P(Z_j \mid \mathbf{u})$ for all $j$, up to the state level. This identification is based on the relative ordering of the probabilities and ensures that we can recover the marginal distributions of all independent latent variables. $\qquad\square$

### A.4 EXPERIMENTAL DETAILS

When dealing with the **Big Five** and **NASDAQ-listed stocks** datasets, we attach distinct domains to samples and discard the domains with sample size less than 20. We then randomly initialize free parameters and fit the model by maximum likelihood.

### A.4.1 ADDITIONAL RESULTS

We present additional results on simulated datasets on different values of latent variables, which further validate both local and strict identifiability. For local identifiability, entries with values less than 0.1, indicating empirical identifiability, are shaded. Additionally, for each $D$ and $N$, we underline the entries corresponding to the minimum number of theoretically required domains. These results demonstrate strong empirical alignment with our theoretical conclusions. For strict identifiability, we observe that when the latent variables are independent, the number of required domains decreases significantly.

| $K/N$ | 2 | 3 | 4 | 5 |
|---|---|---|---|---|
| 1 | 0.2638 | 0.1663 | 0.2484 | 0.1596 |
| 2 | 0.2818 | 0.2758 | 0.1575 | 0.1019 |
| 3 | 0.2252 | 0.2504 | 0.1249 | 4.193e-02 |
| 4 | 0.2505 | 0.1664 | 6.708e-02 | 6.421e-03 |
| 5 | 0.2215 | 0.2210 | 4.984e-02 | 3.830e-03 |
| 6 | 0.1855 | 9.428e-02 | 2.703e-02 | 5.291e-06 |

Table 5: Local Identifiability: $KL_{\mathbf{X}|\mathbf{Z}}$ with $D = 4$

| $K$/$N$ | 2 | 3 | 4 | 5 |
|---|---|---|---|---|
| 1 | 0.4583 | 0.8722 | 0.1796 | 3.786e-06 |
| 2 | 0.4217 | 0.4589 | 7.671e-07 | 1.404e-07 |
| 3 | 0.3227 | 0.0371 | 2.097e-06 | 5.839e-07 |
| 4 | 0.1430 | 7.567e-04 | 2.917e-04 | 4.056e-08 |
| 5 | 0.2266 | 5.462e-08 | 1.235e-06 | 1.050e-06 |
| 6 | 0.1180 | 2.099e-06 | 1.080e-06 | 1.981e-06 |

Table 6: Local Identifiability: $KL_{\mathbf{Z}|\mathbf{u}}$ with $D = 2$

| $K$/$N$ | 2 | | 3 | | 4 | |
|---|---|---|---|---|---|---|
| | 100 | 10.27 | 3 | 23.44 | 2 | 0.1345 |
| - | - | - | 4 | 8.146e-05 | 3 | 3.934e-05 |

Table 7: Strict Identifiability: $KL_{\mathbf{Z}|\mathbf{u}}$ with $D = 2$

| $K$/$N$ | 2 | | 3 | | 4 | |
|---|---|---|---|---|---|---|
| | 100 | 22.17 | 3 | 4.703 | 2 | 3.321e-02 |
| - | - | - | 4 | 6.307e-06 | 3 | 3.465e-05 |

Table 8: Strict Identifiability: $KL_{\mathbf{X}|\mathbf{Z}}$ with $D = 2$

| $K$/$N$ | 2 | 3 | 4 | 5 |
|---|---|---|---|---|
| 1 | 0.5665 | 2.741e-02 | 0.2620 | 0.2308 |
| 2 | 0.4457 | 0.5006 | 0.1363 | 6.695e-02 |
| 3 | 0.3679 | 0.2747 | 0.1796 | 4.338e-02 |
| 4 | 0.3575 | 0.3372 | 0.1131 | 5.667e-03 |
| 5 | 0.2182 | 0.3333 | 8.595e-02 | 1.059e-02 |
| 6 | 0.3416 | 0.2213 | 6.732e-02 | 1.583e-05 |

Table 9: Local Identifiability: $KL_{\mathbf{Z}|\mathbf{u}}$ with $D = 4$

| $K$/$N$ | 2 | 3 | 4 | 5 |
|---|---|---|---|---|
| 1 | 0.3674 | 0.2450 | 0.2570 | 1.384e-06 |
| 2 | 0.3039 | 0.0745 | 3.898e-07 | 1.007e-07 |
| 3 | 0.1311 | 9.323e-04 | 3.109e-06 | 8.951e-08 |
| 4 | 0.1151 | 5.063e-05 | 2.705e-05 | 7.027e-08 |
| 5 | 0.3204 | 5.566e-08 | 1.545e-07 | 1.047e-06 |
| 6 | 0.0218 | 1.428e-06 | 1.048e-06 | 2.315e-06 |

Table 10: Local Identifiability: $KL_{\mathbf{X}|\mathbf{Z}}$ with $D = 2$

| $K$/$N$ | 6 | | 10 | |
|---|---|---|---|---|
| | 6 | 1.606 | 3 | 0.7397 |
| - | 7 | 5.647e-04 | 4 | 1.452e-05 |

Table 11: Strict Identifiability: $KL_{\mathbf{Z}|\mathbf{u}}$ with $D = 4$

