# OpenReview forum: "Learning Discrete Latent Models from Discrete Observations"
_ICLR.cc/2025/Conference — Submitted to ICLR 2025_

### Official Review · Reviewer_oKQy · 2024-10-19

**Soundness:** 3
**Presentation:** 2
**Contribution:** 3
**Rating:** 5
**Confidence:** 2

**Summary:**

This paper establishes a theoretical foundation for scenarios involving both discrete latent variables and observations. The author(s) discuss one-to-one and flexible mappings, as well as local and strict identifiability under flexible mappings. The results of experiments conducted on both simulated and real-world datasets confirm the required number of domains of identifiability as derived from the theorems.

**Strengths:**

1. The author's use of multiple domains relaxes certain conditions, enabling flexible modeling without imposing restrictive assumptions.
2. It identifies marginal distributions based on specific assumptions.

**Weaknesses:**

1. I find that Assumption 1.2 appears to be too general. For instance, in the example illustrated in Figure 1, it's possible that X3 could be a cause of X1.
2. I'm not an expert in ICA and found myself waiting for an explanation of both ICA and the term `domain` as I read. It would be helpful for the author to inform readers early on that Appendix A1 contains the necessary background knowledge.

**Questions:**

1. For the abstract, providing the full name of ICA could be better.
2. For assumption 1.1, should it be P(X,Z|k)=P(X|Z)p(Z|k)?
3. For Theorem 2, you said after taking the log on both sides of 2. What is 2? And why do the linear equations have unique solutions if and only if $2^{D-1} \ge D$?
4. For line 652, please add more examples.
5. For line 231-232, why for $\alpha$, the l ranges to $2^D-1$?

---

> ### Author Response · Authors · 2024-11-24
>
> We sincerely appreciate the valuable reviews and constructive comments, which helped improve the quality of this paper. We have carefully reviewed the manuscript as per your suggestions. We have revised the paper accordingly, and below, we provide detailed responses to your comments.
>
> >***W1***: I find that Assumption 1.2 appears to be too general. For instance, in the example illustrated in Figure 1, it's possible that X3 could be a cause of X1.
>
> - ***A1**: Thank you for raising this concern. Assumption 1.2 is indeed practical and aligns with many real-world applications. For instance, as mentioned in the introduction, it holds in contexts such as survey data in social sciences and topic modeling in natural language processing. Figure 1 provides a concrete example to illustrate this point. While it may appear that  $X_3$  and  $X_1$ are dependent or causally related, it is reasonable to conclude that such relationship becomes insignificant when the disease is observed. In this case,  $X_3$  does not serve as a significant cause of  $X_1$.
>
>
>
> >***W2***: I'm not an expert in ICA and found myself waiting for an explanation of both ICA and the term `domain` as I read. It would be helpful for the author to inform readers early on that Appendix A1 contains the necessary background knowledge.
>
> - ***A2***: Thank you for your suggestion. We have clarified the term “domain” in the paper by adding the following explanation: Here, “multi-domain” refers to distinct scenarios or conditions under which the distributions of variables differ.
>
> For ICA, we have added a a short explanation of what ICA is in the main text and a reference to the appendix where ICA is discussed in greater detail, allowing readers with limited background to gain a clearer understanding: For instance, in ICA, the observed data is considered a mixture of independent, unobserved components, and the goal is to "demix" these observations to recover the latent variables, up to some ambiguity (please refer to Appendix A.1.1 for more details).
>
> >***Q1***: For the abstract, providing the full name of ICA could be better.
> - ***A1***: Thanks for the advice, we have modified in the paper: In approaches where certain uniqueness of representation is guaranteed, such as nonlinear Independent Component Analysis
>
> >***Q2***: For assumption 1.1, should it be $P(X,Z\mid k)=P(X\mid Z)p(Z\mid k)$?
> - ***A2***: Thanks for the correction of this typo. We have corrected it as follows: Across all domains, the binary observed variables $\mathbf{X} = \{X_1, \cdots, X_N\}$ are generated from the binary latent variables $\mathbf{Z} = \{Z_1, \cdots, Z_D\}$ according to the factorization $P(\mathbf{X} \mid \mathbf{u}) = \sum_{l=1}^{2^D}P(\mathbf{X} \mid \mathbf{Z}=\mathbf{z}_l)P(\mathbf{Z} =\mathbf{z}_l \mid \mathbf{u})$.
>
>
> >***Q3***: For Theorem 2, you said after taking the log on both sides of 2. What is 2? And why do the linear equations have unique solutions if and only if $2^{D-1} \ge D$
>
> - ***A3***: Thank you for pointing this out. The “2” refers to Equation 2—apologies for the confusion. We have clarified this in the paper. After taking the logarithm of the equations, they are transformed into linear equations. The identification of parameters in these linear equations depends solely on the relationship between the number of equations and the number of unknowns. According to Equation 2, the number of equations is $2^{D-1}$, while the number of unknowns is D, as we assume the binary latent variables are independent of each other. Therefore, identifiability is established if and only if $2^{D-1} \geq D$.
>
> >***Q4***: For line 652, please add more examples.
>
> - ***A4***: Thank you for pointing this out. We have updated the paper to include the following revision:: Domain information is widely observed in the real world, such as labels across populations, different experimental conditions, varying time periods, geographic regions, or demographic groups (e.g., age, gender, or socioeconomic status). These distinctions are often integral to many datasets, rendering the theorem adaptable to various downstream problems.
>
>
> >***Q5***: For line 231-232, why for $\alpha$, the l ranges to $2^D - 1$?
>
> - ***A5***: Thank you for the question. For line 231-232, we assume that the latent variables in the model can be dependent. In that case, the free parameters $\alpha$ for characterizing the marginal distributions of latent variables ($\mathbf{Z}$) in each domain have the total number of $2^D - 1$. For better clarification, we have revised the content as follows: Based on Equation 1, we introduce additional notations to better represent the free parameters we aim to identify. Let $ \alpha_{l,k} = P(\mathbf{Z} = \mathbf{z}_l \mid k) $, where $ l \in (\{1, \dots, 2^D - 1\} )$, represent the free parameters used to characterize the marginal probabilities of the latent variables.

---

> > ### Comment · Reviewer_oKQy · 2024-11-25
> >
> > Thanks for addressing all my questions in detail. I'd like to keep my scores, but from the point of view of other reviewers, it would be better to rewrite it in a more explicit version.

---

> > > ### Author Response · Authors · 2024-11-27
> > >
> > > Thank you for your recognition. We have updated the manuscript and please let us know if you have any additional comments or questions.

---

### Official Review · Reviewer_hqQM · 2024-10-24

**Soundness:** 3
**Presentation:** 3
**Contribution:** 3
**Rating:** 6
**Confidence:** 3

**Summary:**

This paper studies mathematical conditions to achieve identifiability in discrete-discrete latent variable models. I think this is an interesting paper with good theoretical results. I am happy to raise my score should the authors respond thoughtfully to my comments and questions.

**Strengths:**

- Clear presentation
- Closes a clear gap in the literature
- Relevant overall topic (latent variable models in representation learning)

**Weaknesses:**

- The empirical examples are both not really discrete but rather discretized, which of course is not ideal. Since the paper is primarily theoretical and just has some empirical examples as a “nice addition”, I think this is fine.
- Still I would like to see real examples, where we *actually* have both discrete observed and discrete latent variables. The authors argue with the cases where we would, for example, discretize personality traits, which as a psychologist by training originally, I have to reject as not a good (oversimplifying) approach.

**Questions:**

- The authors often state the inequalities involving N, K, D as weak, e.g Assumption 4.1 and similar. Can the authors explain a bit more on why they think these are weak? I.e. can we expect these to be justified in most real world cases?
- Can the authors comments on the major difference between Theorem 3 and 4. I.e. in which way does the independence assumption relax the other requirements?
- Results simulation study: When the authors say that they show empirically that the assumptions of Theorem 6 can be strongly weakend, can they say which idendifiablilty constraints the theorem would imply in this concrete case (3 latent variables, 7 or 8 observed variables)?

---

> ### Author Response · Authors · 2024-11-24
>
> We sincerely appreciate your insightful and constructive feedback, which has greatly enhanced the clarity of our theories and models, as well as the robustness of our experiments. In response, we have updated the paper accordingly. Below, please find our detailed responses to your comments.
>
>
> >***W1***: The empirical examples are both not really discrete but rather discretized, which of course is not ideal. Since the paper is primarily theoretical and just has some empirical examples as a “nice addition”, I think this is fine.
>
> - ***A1***: Thank you for pointing this out. For the two empirical examples, the original observed variables from ****Big Five****  are discrete, as they measure five levels of agreement. In contrast, the ****NASDAQ-listed**** dataset is continuous. Since real-world data can often be inherently discrete or discretized during collection or preprocessing, we manually discretized this dataset to fit the scope of our study. The goal of this section is to demonstrate that regardless of the data’s generating process—whether naturally discrete or discretized—we can reliably recover the latent variable distributions and uncover meaningful relationships between latent and observed variables. This is achieved without imposing strict constraints on the data generation process, demonstrating the robustness of our approach.
>
> >***W2***: Still I would like to see real examples, where we *actually* have both discrete observed and discrete latent variables. The authors argue with the cases where we would, for example, discretize personality traits, which as a psychologist by training originally, I have to reject as not a good (oversimplifying) approach.
>
> - ***A2***: Thank you for your thoughtful comment and we understand the concern. For the personality traits case, the observed variables are categorical, as they are measured on a scale with five levels of agreement. There are indeed many real-world applications where both observed and latent variables are discrete. For instance, as shown in Figure 1, in medicine, symptoms (observed) and potential diseases (latent) are frequently represented as binary variables. This highlights the practical importance of our approach in scenarios involving discrete data. In social sciences, survey data frequently involve discrete observed and latent variables, such as levels of satisfaction for questions and underlying respondent traits. Another concrete example from social science is the Trends in Mathematics and Science Study (TIMSS), an international, large-scale assessment that evaluates the mathematics skills and science knowledge of students across different grades. The variables in this dataset are binary and are widely analyzed using latent class models.
>
>
> >***W3***: The authors often state the inequalities involving N, K, D as weak, e.g Assumption 4.1 and similar. Can the authors explain a bit more on why they think these are weak? I.e. can we expect these to be justified in most real world cases?
>
> - ***A3***: Thank you for your question. To clarify, in the inequality constraints,  N  represents the number of observed variables,  K  represents the number of domains, and  D  represents the number of latent variables.
>
> - The assumptions we make are relatively mild, as many datasets typically have a sufficient number of observed variables. However, we acknowledge that the applicability of these conditions depends on the specific data. For instance, if the number of observed variables is limited, achieving the sufficient conditions for identifiability may require a large number of domains (potentially even as many as the observed variables). That said, under alternative settings—such as when the latent variables are assumed to be independent—the conditions may be relaxed.
>
> - We also recognize that there are real-world scenarios where the sufficient conditions may not hold. For example, when there is only one observed variable in just one domain, the model is certainly unidentifiable. Addressing such cases and generalizing our current framework is an important direction for future work, and we are keen to explore this in subsequent research.

---

> > ### Author Response · Authors · 2024-11-24
> >
> > >***Q2***: Can the authors comments on the major difference between Theorem 3 and 4. I.e. in which way does the independence assumption relax the other requirements?
> >
> > - ***A2***: Thank you for your question. The key difference between the settings underlying Theorem 3 and Theorem 4 lies in whether the independence assumption is imposed on the latent variables. When we assume independence between latent variables within each domain, the total number of unknown parameters is significantly reduced. For instance, if we have 10 binary latent variables and 1 domain, their marginal distributions would typically require $ 2^{10} - 1 $ free parameters to characterize. However, under the independence assumption, this reduces to just 10 free parameters. By reducing the number of free parameters, the sufficient conditions for local identifiability (as outlined in Theorems 3 and 4) are less restrictive and can be relaxed. We hope this explanation clarifies the distinction, and we are happy to provide further details if needed.
> >
> > >***Q3***: Results simulation study: When the authors say that they show empirically that the assumptions of Theorem 6 can be strongly weakend, can they say which idendifiablilty constraints the theorem would imply in this concrete case (3 latent variables, 7 or 8 observed variables)?
> >
> >
> > - ***A3***: Thank you for your question. In Theorem 6, we stated the sufficient condition for identifiability as $N \geq 2^D $ and $ K \geq 2^{(D+1)} - N $. However, our simulation studies demonstrate that the number of required domains in practice is often much smaller than this theoretical bound. For instance, with 3 latent variables and 8 observed variables, strict identifiability can be achieved with just 2 domains, instead of the 8 domains suggested by the sufficient condition.
> >
> >
> > ***References***
> >
> > [1]Martin, M. O., Mullis, I. V. S., & Hooper, M. (Eds.). (2016). TIMSS 2015 International Results in Science. International Association for the Evaluation of Educational Achievement (IEA). Retrieved from https://timssandpirls.bc.edu/timss2015/international-results/
> > [2]Goldberg, L. R. (1992). The development of markers for the Big-Five factor structure. Psychological Assessment, 4(1), 26–42. https://doi.org/10.1037/1040-3590.4.1.26

---

### Official Review · Reviewer_FV1R · 2024-10-30

**Soundness:** 2
**Presentation:** 1
**Contribution:** 2
**Rating:** 3
**Confidence:** 3

**Summary:**

This paper investigates the problem of identifying meaningful representations in high-dimensional data, a central objective in representation learning. While some methods offer unique representations, such as nonlinear independent component analysis (ICA), these approaches generally assume continuous variables, with recent work extending identifiability to binarized observed variables. However, there has been no established framework for cases involving discrete latent variables. In response, the authors propose a method to leverage multi-domain information for achieving identifiability when both latent and observed variables are discrete. They present general identification conditions that do not rely on specific data distributions or parametric models. Experimental results, conducted on both simulated and real-world datasets, are used to validate the effectiveness of the proposed approach.

**Strengths:**

To the best of my knowledge, the paper studies a scenario (discrete observed variables + discrete latent variables) that has not been explored so far.

**Weaknesses:**

I find the paper extremely unclear in terms of motivation, intuition behind results/assumptions, and overall presentation. This severely hinders the readability of the article and the credibility of its claims. Here are some more specific concerns of mine:

1. The introduction of $K$ domains is not justified. What are these domains modeling? Why are they necessary in the analysis?

2. Definition 1 is not precise, it needs to be mathematized in order to be relied on to follow proofs;

3. The proof of Theorem 3 is unclear. Also, assuming that the proof is correct, given how trivial it is, does this result really deserve the name of "Theorem"?

4. The definition of local identifiability needs to be (1) mathematized, (2) moved to the main text, as it seems pretty central to the paper's focus;

5. The explanation after Theorem 3 is unsatisfactory (e.g., why is it ok to assume that the columns of the Jacobian are linearly independent?);

6. In Lemma 2, the two $\otimes, \tilde \otimes$ notations are unclearly differentiated;

7. The experiments, especially the real world ones, do not seem to match the goals of the paper. One of the points made at the beginning is that, in some applications, it may be crucial to model latent variables as discrete rather than continuous, especially when they may have a certain interpretation for the science domain in question. However, here the authors don't really justify their choice of data through this lens (at some point, they even discretize originally continuous data) nor do they compare the performance of their method with continuous latent variable alternatives. This casts doubt on the ability of these experiments to validate the author's claims;

8. More broadly, while it may be true that no previous work looked at this categorical-categorical setting, I wonder if this is because such a setting has no peculiarity of its own. Indeed, looking at the proofs, no new interesting technique seems to have been introduced. Nevertheless, I would be happy to hear what the authors may have to say to disprove this feeling of mine.

**Questions:**

As I mentioned above, the paper has many unclear points (at least to me). Here are some questions that could help the authors clarify those points:

1. In Assumption 2, do the authors mean for all $k$?

2. In Assumption 4, why is it reasonable to assume that the derivatives wrt all free parameters are positive?

3. In Definition 2, wouldn't a $2^D$ vector suffice (once you know $P(X_i = 0 \mid \boldsymbol Z = \boldsymbol z_l)$, you also know $P(X_i = 1 \mid \boldsymbol Z = \boldsymbol z_l) = 1 - P(X_i = 0 \mid \boldsymbol Z = \boldsymbol z_l)$)?

---

> ### Author Response · Authors · 2024-11-24
>
> Thank you for your insightful and constructive feedback, which indeed improves the readability of our theories and models as well as the soundness of our experiments. We provide the point-to-point response to your comments below and have updated the paper accordingly. Please find our point-by-point responses below.
>
>
> >***W1***: The introduction of K domains is not justified. What are these domains modeling? Why are they necessary in the analysis?
>
> - ***A1***: Thank you for your thoughtful questions. The multi-domain setup is an inherent property of our model, meaning that the distributions of latent variables depend on an auxiliary variable as part of the model generation process. We utilize this property in establishing the identifiability of the model.
>
> - The motivation for introducing multi-domain settings in our work is as follows: Most, if not all, previous studies on identifiability in multi-domain scenarios assume that the variables are continuous. In contrast, we are the first to establish identifiability in the discrete case and demonstrate how multi-domain information can significantly relax the parameter constraints that would otherwise be more restrictive in single-domain settings.
>
> - For example, assuming independent latent variables, when the number of latent variables is 3 $(D=3)$ and the number of observed variables is also 3 $(N=3)$, the distributions are not locally identifiable, as demonstrated by the simulation results in Table 1. However, if we have 6 or more domains, we can effectively identify all distributions, even when only 3 observed variables are available per domain.
>
> - In real-world applications, obtaining multiple domains is often practical and feasible. For instance, participants’ demographic information in survey data, user information in recommender systems, and object categories in image or video analysis are examples where multi-domain information is readily available and can be leveraged.
>
> >***W2***: Definition 1 is not precise, it needs to be mathematized in order to be relied on to follow proofs;
>
> - ***A1***: Thanks for your suggestion, we have revised that in the paper as follows:
> - Let $ X $ be a random variable with a discrete set of possible states $ \{s_1, s_2, \dots, s_k\} $ and an associated probability distribution $ P(X = s_i) = p_i $ for $ i = 1, \dots, k $, where $ p_i \geq 0 $ and $ \sum_{i=1}^k p_i = 1 $. The random variable $ X $ is said to have ***state-level identifiability*** if its distribution is identifiable up to a permutation of its states.
> Formally, this means there exists a permutation $ \sigma \in S_k $ such that: $P_Y = [\{p_{\sigma(1)}, p_{\sigma(2)}, \dots, p_{\sigma(k)}\}$], where $ \sigma $ is an element of the symmetric group $ S_k $, representing all possible permutations of $ \{1, 2, \dots, k\} $. In other words, the specific labeling of the states $ \{s_1, s_2, \dots, s_k\} $ may not be recoverable, but the probability distribution over the states $ \{p_1, p_2, \dots, p_k\} $ is uniquely determined up to a permutation of the states.
>
> >***W3***: The proof of Theorem 3 is unclear. Also, assuming that the proof is correct, given how trivial it is, does this result really deserve the name of "Theorem"?
>
> - ***A1***: Thank you for pointing this out. I assume you are referring to Theorem 2, as the proof of Theorem 3 is quite comprehensive in the appendix. Please feel free to correct me if this is not what you intended. For Theorem 2, we hope you agree that it is more appropriate to use “Lemma” instead of “Theorem” in this context. We have updated the manuscript accordingly.
>
> >***W4***: The definition of local identifiability needs to be (1) mathematized, (2) moved to the main text, as it seems pretty central to the paper's focus;
>
> - ***A4***: Thank you for your suggestion, we have mathematized it and moved it into the main text in the updated manuscript:
>
> - Let $ \mathcal{M}(\theta) $ denote a parametric model, where $ \theta \in \Theta $ represents the parameter vector in the parameter space $ \Theta $, and $ \mathcal{M}(\theta) $ maps the parameter $ \theta $ to the set of observed data distributions.
> The model $ \mathcal{M}(\theta) $ is \textbf{locally identifiable} at $ \theta_0 \in \Theta $ if: $\exists \, \epsilon > 0 \text{ such that } \forall \theta \in B_\epsilon(\theta_0) \cap \Theta, \quad \mathcal{M}(\theta) = \mathcal{M}(\theta_0) \implies \theta = \theta_0,$
> where: $ B_\epsilon(\theta_0) = \{ \theta \in \Theta : \|\theta - \theta_0\| < \epsilon \} $ is the open ball of radius $ \epsilon $ centered at $ \theta_0 $, $ \|\cdot\| $ denotes a norm (e.g., Euclidean norm) on the parameter space $ \Theta $, $ \mathcal{M}(\theta) $ is the mapping from parameters $ \theta $ to the observed data distributions.

---

> > ### Author Response · Authors · 2024-11-24
> >
> > >***W5***: The explanation after Theorem 3 is unsatisfactory (e.g., why is it ok to assume that the columns of the Jacobian are linearly independent?);
> >
> > - ***A5***: Thank you for the question. The assumption that the Jacobian matrix having full column rank suggests that no column can be expressed as a linear combination of others. This suggests that changes in the input variables produce independent directions of change in the output. Namely, there is no exact ill-posed coupled cases when the change from different variables are the same. This is actually a quite natural assumption, similar to rank faithfulness assumption where rank faithfully represents the causal or structure relations without accidental cancenllations (Peters et al., 2017; Hyvärinen et al., 2010). If the parameters are randomly chosen, the chance for them to violate such assumption is very slim.
> >
> >
> >
> > >***W6***: In Lemma 2, the two notations are unclearly differentiated;
> >
> > - ***A6***: Thank you for the suggestion. We have added more explanations of these symbols in the updated manuscript: $\otimes$ denotes the outer product of vectors and $\tilde{\otimes}$ denotes the outer product between a tensor and a vector
> >
> >
> > >***W7***: The experiments, especially the real world ones, do not seem to match the goals of the paper. One of the points made at the beginning is that, in some applications, it may be crucial to model latent variables as discrete rather than continuous, especially when they may have a certain interpretation for the science domain in question. However, here the authors don't really justify their choice of data through this lens (at some point, they even discretize originally continuous data) nor do they compare the performance of their method with continuous latent variable alternatives. This casts doubt on the ability of these experiments to validate the author's claims;
> >
> >
> > - ***A7***: Thank you for your suggestions. To the best of our knowledge, there are no existing methods applicable for comparison in this setting, as none provide guarantees of identifiability in the discrete case. Moreover, the primary goal of our experiments is to validate the identifiability of our theoretical results rather than to propose a new estimation method. We believe that demonstrating how our experiments verify these identifiability results is sufficient to address this objective.
> >
> > - In our experiments, we aim to demonstrate that our methods are applicable across a wide range of scenarios. In practice, data is often discrete, either inherently (e.g., the Big Five dataset) or through discretization from continuous variables (e.g., the NASDAQ dataset). By including both types of data, we show that our results are robust and applicable regardless of whether the data is inherently discrete or discretized from continuous variables.

---

> > > ### Author Response · Authors · 2024-11-24
> > >
> > > >***W8***: More broadly, while it may be true that no previous work looked at this categorical-categorical setting, I wonder if this is because such a setting has no peculiarity of its own. Indeed, looking at the proofs, no new interesting technique seems to have been introduced. Nevertheless, I would be happy to hear what the authors may have to say to disprove this feeling of mine.
> > >
> > > - ***A8***: Thanks for your concern on the possibility of limited applications of our model. While the discrete-to-discrete setting may seem peculiar, there are numerous real-world problems where our methods can provide valuable solutions. As mentioned in the introduction, in medicine, symptoms (observed) and potential diseases (latent) are often represented as binary variables. Similarly, whether a patient receives treatment and recovers can also be modeled using binary variables. In social sciences, survey data frequently involve discrete observed and latent variables, such as levels of satisfaction for questions and underlying respondent traits. One concrete dataset in social science is the  Trends in Mathematics and Science Study (TIMSS) is an international and large-scale assessment to evaluate the mathematics skills
> > > and science knowledge of students in different grades, which is also widely analyzied in latent class models. Another example is topic modeling in natural language processing (NLP), where words or phrases in documents are encoded as discrete variables, and the unobserved topics are also modeled as discrete latent variables. These examples highlight the practical relevance of our approach across various domains.
> > >
> > >
> > > - In terms of your concern on the technical side, this paper primarily leverages techniques such as linear algebra, factor analysis, and N-way array decomposition, all of which are well-established in the literature (Oseledets et al., 2009; Shapiro, 1985; Wegge, 1996; Sidiropoulos & Bro, 2000). The goal of the paper is not to develop entirely new mathematical techniques but to address a novel discrete-to-discrete setting, which is both practically important and largely unexplored. We respectfully disagree with the notion that using simple techniques is a shortcoming. On the contrary, we view this as an advantage. Tackling critical problems with simple assumptions and accessible tools makes the work more impactful and widely applicable. Many influential works that have significantly advanced real-world problems also rely on straightforward techniques. For example, foundational results on nonlinear ICA (Hyvärinen \& Sasaki, 2019; Khemakhem et al., 2020; Zheng et al., 2022) use relatively simple tools such as calculus, linear algebra, and probabilistic modeling to establish identifiability. Similarly, our use of simple but effective techniques provides a clear and reliable framework to address the discrete-to-discrete setting.
> > >
> > >
> > > ***References***
> > >
> > > [1]Oseledets, I. V., Savostyanov, D. V., & Tyrtyshnikov, E. E. (2009). Linear algebra for tensor problems. Computing, 85(3–4), 169–188. https://doi.org/10.1007/s00607-009-0047-0
> > > [2]Shapiro, A. (1985). Identifiability of factor analysis: Some results and open problems. Linear Algebra and Its Applications, 70, 1–7. https://doi.org/10.1016/0024-3795(85)90123-6
> > > [3]Wegge, L. L. (1996). Local identifiability of the factor analysis and measurement error model parameter. Journal of Econometrics, 70(2), 351–382. https://doi.org/10.1016/0304-4076(95)01791-7
> > > [4]Sidiropoulos, N. D., & Bro, R. (2000). On the uniqueness of multilinear decomposition of N-way arrays. Journal of Chemometrics: A Journal of the Chemometrics Society, 14(3), 229–239. https://doi.org/10.1002/1099-128X(200005/06)14:3<229::AID-CEM589>3.0.CO;2-U
> > > [5]Hyvärinen, A., & Sasaki, H. (2019). Nonlinear ICA using auxiliary variables and generalized contrastive learning. Proceedings of the 22nd International Conference on Artificial Intelligence and Statistics (AISTATS), 89, 859–868. http://proceedings.mlr.press/v89/hyvarinen19a.html
> > > [6]Khemakhem, I., Kingma, D. P., Monti, R., & Hyvärinen, A. (2020). Variational autoencoders and nonlinear ICA: A unifying framework. Proceedings of the 23rd International Conference on Artificial Intelligence and Statistics (AISTATS), 108, 2207–2217. http://proceedings.mlr.press/v108/khemakhem20a.html
> > > [7]Zheng, Y., Ng, I., & Zhang, K. (2022). On the identifiability of nonlinear ICA: Sparsity and beyond. Advances in Neural Information Processing Systems (NeurIPS), 35, 18957–18969. https://proceedings.neurips.cc/paper_files/paper/2022/hash/476ea3a0f3c4225f95c98ca7db67cfdb-Abstract-Conference.html

---

> > > > ### Author Response · Authors · 2024-11-24
> > > >
> > > > >***Q1***: In Assumption 2, do the authors mean for all $k$?
> > > > - ***A1***: Thanks for the question. Yes, this assumption should hold for any domain.
> > > >
> > > > >***Q2***: In Assumption 4, why is it reasonable to assume that the derivatives wrt all free parameters are positive?
> > > > - ***A2***: Thank you for the question. Assumption 4 states that “The free parameters are all positive”; This does not imply anything about the signs of their derivatives or the entries of the Jacobian matrix.
> > > >
> > > > >***Q2***: In Definition 2, wouldn't a $2^D$ vector suffice?
> > > > - ***A2***: Thank you for the question. You are correct that the free parameter is singular for $P(\mathbf{X}_i \mid \mathbf{z}_l) $. In our construction, we utilize N-way array decomposition techniques, where $ P(\mathbf{X} \mid \mathbf{u}) $ (with $ \mathbf{u} $ now the updated domain variable) is decomposed into submatrices. The matrix in Definition 2 serves as a notation to better represent this decomposition process, rather than identifying the free parameters in the model. Please feel free to let us know if there is any further confusion.

---

> > > > ### Comment · Reviewer_FV1R · 2024-11-25
> > > >
> > > > I thank the authors for carefully responding to my comments. I am quite positive that, after implementing the promised changes and a *major* rewriting of the paper to enhance its clarity, the article will be ready for publication at a top venue. However, I do not feel comfortable raising my score as I do think that there is major work to be done before the paper is in good enough shape -- I think the amount of such work makes it inappropriate to possibly lead to publication without further scrutiny from peer reviewers, and therefore I do not raise my score.
> > > >
> > > > P.S. I agree with the authors that simplicity of the mathematical arguments is a plus, not a minus. What I was trying to provocatively push the authors to discuss was why, from a technical point of view, the treatment of this new setting deserves an altogether new article -- without further discussion of this point (which I encourage them to add in the next version of the paper) and looking at the proofs, one could otherwise argue that the results presented here are an (almost) obvious consequence of previous results or of (almost) trivial math.

---

> > > > > ### Author Response · Authors · 2024-11-27
> > > > >
> > > > > Thank you for your thoughtful comment and recognition. We have updated the manuscript, particularly emphasizing the significance and uniqueness of this work. Please let us know if you have any additional comments or suggestions.

---

### Official Review · Reviewer_X6ob · 2024-11-02

**Soundness:** 3
**Presentation:** 1
**Contribution:** 2
**Rating:** 3
**Confidence:** 3

**Summary:**

The paper presents a set of conditions for model identifiability in the binary data case. The authors have developed separate conditions for both local and global identifiability. The paper also addresses identifiability under very strict assumptions regarding the mapping between latent factors and observed variables. While similar results exist under similar frameworks, the authors' multi-domain approach represents a novel contribution to the field. The authors also conduct a set of experiments designed to prove the paper's main claims.

**Strengths:**

* The paper tackles an important issue, as model identifiability is crucial.
* The paper's introduction is well-constructed, explaining concepts clearly and providing appropriate illustrations
* The problem is well-defined, and the derivations appear correct under the stated assumptions

**Weaknesses:**

* While the section ordering is logical, the paper's writing is confusing, and lacks a well-defined summary of conditions for each type of identifiability. Some assumptions are stated redundantly (e.g., latent factors being mutually independent within each domain, positive free parameters), and some results appear unnecessary, adding to the confusion.

* The assumptions, which are central to the paper as they constitute the conditions for identifiability, require more explanation and discussion of their practicality:
  - Assumption 1.2 is unrealistic in medicine, for example (although admittedly common in the field)
  - Assumption 2 creates a degenerate model as described earlier
  - Assumptions 4.2 and 5.2 appear particularly rushed and unclear; for instance, would a redundant (always 1) observed variable break model identifiability?
  - Assumption 6, which is core to the proof, is almost identical to the target proposition, making it seemingly circular


* The paper claims in the abstract: 'In this paper, we show how multidomain information can be leveraged to achieve identifiability.' However, it's not clear from the paper how the multidomain setup "leveraged" identifiability. Moreover, given that identifiability is a model property, it's unclear how a 'multidomain' setup can be 'leveraged.' Furthermore, the need for a new work for the multi-domain setup isn't well explained, which is problematic given this is the paper's claimed novelty.

* There is insufficient explanation of how the provided experiments prove the claimed identifiability.
While the paper derives sufficient conditions for identifiability, there is no discussion about whether these conditions are necessary and to what extent, thus limiting the paper's contribution.

* The paper's title focuses on discrete distributions, although the content appears to address only binary distributions.

* Section 3 appears redundant as no other sections rely on it, and its results are trivial under such strict assumptions, leading to a degenerate model. The same applies to Theorem 2.

**Questions:**

* Lines 036-038: Citation needed?

* Lines 050-051, '...deep learning techniques to ensure identifiability': This requires clarification, as identifiability is a model property rather than an algorithmic one. How can deep neural networks 'ensure' identifiability?

---

> ### Author Response · Authors · 2024-11-24
>
> Thank you for your insightful and constructive feedback, which indeed improves the readability of our theories and models as well as the soundness of our experiments. We provide the point-to-point response to your comments below and have updated the paper accordingly.
>
> With gratitude, we have included a new section summarizing the conditions for each type of identifiability. We have also added more explanations or details to the assumptions and experiments. The manuscript has been thoroughly revised in light of your constructive suggestions. Please find our point-by-point responses below.
>
> >****W1:**** While the section ordering is logical, the paper's writing is confusing, and lacks a well-defined summary of conditions for each type of identifiability. Some assumptions are stated redundantly (e.g., latent factors being mutually independent within each domain, positive free parameters), and some results appear unnecessary, adding to the confusion.
>
> - ***A1:1*** This is a great suggestion, below please see a summary of conditions and a combination of conditions for each type of identifiability. It is also updated in the manuscript.
> 1. One-to-one mapping + Assumption [1], [2]: strict identifiability on $P(X_i\mid \mathbf{Z})$ and $P(\mathbf{Z}\mid \mathbf{u})$.
> 2. One-to-one mapping + Assumption [1], [2], [3]: strict identifiability on $P(X_i\mid \mathbf{Z})$ and $P(Z_j \mid \mathbf{u})$.
> 3. Flexible mapping + Assumption [1], [4]: local identifiability on $P(X_i\mid \mathbf{Z})$ and $P(\mathbf{Z}\mid \mathbf{u})$.
> 4. Flexible mapping + Assumption [1], [3], [5]: local identifiability on $P(X_i\mid \mathbf{Z})$ and $P(Z_j \mid \mathbf{u})$.
> 5. Flexible mapping + Assumption [1], [6]: strict identifiability on $P(X_i\mid \mathbf{Z})$ and $P(\mathbf{Z}\mid \mathbf{u})$.
> 6. Flexible mapping + Assumption [1], [3],[6],[7]: strict identifiability on $P(X_i\mid \mathbf{Z})$ and $P(Z_j \mid \mathbf{u})$.
> - ***A1.2*** Thank you for your suggestions regarding the paper’s structure. In our view, it was necessary to state certain assumptions, such as those concerning positive free parameters, separately for different theorems because the constraints apply to distinct parameters across them. To address potential redundancy, we highlighted the differences in the assumptions involved in the various results. For example, in Assumption 5, we replaced "The free parameters are all positive" with: "The free parameters {$\gamma_{(1,1)}, \cdots, \gamma_{(D,K)}, \beta_{(1,1)}, \cdots, \beta_{(N,2^D)}$}are all positive." Hope this can address your concern properly.
>
> >***W2:*** The assumptions, which are central to the paper as they constitute the conditions for identifiability, require more explanation and discussion of their practicality:
>
> >***W2.1***: Assumption 1.2 is unrealistic in medicine, for example (although admittedly common in the field)
>
> - ***A2.1***: We appreciate your observation and acknowledge that, in practice, observed variables may sometimes influence each other. Allowing directed interactions among measured variables would be a very interesting line of future research. In this paper, we primarily focus on scenarios where observed variables are conditionally independent given the latent variables—a setting that applies to many practical situations. For instance, in healthcare, as shown in Figure 1, symptoms are often conditionally independent when conditioned on their underlying diseases.  While it might seem that nausea and fatigue frequently occur together, these symptoms are reflections of an underlying condition, such as gastritis. Thus, they are expected to be conditionally independent when the disease is accounted for. Also, in questionnaire studies, the answers to questions are generally assumed to be conditionally independent given the underying mental cinditions.

---

> > ### Author Response · Authors · 2024-11-24
> >
> > >***W2.2***: Assumption 2 creates a degenerate model as described earlier
> >
> > - ***A2.2***: We understand that the discussion under the invertible mapping assumption may appear trivial. At the same time, note that we aim to find underlying true variables. This invertibility has been widely assumed in continuous case, in which one aims to find the underlying hidden variables from their mixtures in the linear case (for instance, see Hyvärinen et al. (2001); Hyvärinen \& Oja (2000)) or nonlinear cases (see, e.g., Hyvärinen \& Morioka, 2017; Hyvärinen \& Sasaki, 2019; Khemakhem et al., 2020; Zheng et al., 2022). It has been shown that identification of the (deterministic) mixing function is non-trivial. As you see from this manuscript, the same phenomenon happens in the discrete case, and our result in the one-to-one mapping case can be seen as a counterpart of the developments above in the discrete case--without the independence assumption, it will not be possible.
> >
> > Moreover, in the continuous case, the problem is hard to deal with if the mapping from the hidden variables to the measured ones is not one-to-one (for instance, because of noise).  As their counterpart in the discrete case, interestingly, our results show that under suitable assumptions, the whole generating process is identifiable.
> >
> > >***W2.3***: Assumptions 4.2 and 5.2 appear particularly rushed and unclear; for instance, would a redundant (always 1) observed variable break model identifiability?
> >
> > - ***A2.3***: Thank you for raising this point, we have revised Assumption 4.2 and 5.2 to bring more clarity:
> >
> > Assumption 4.2: The free parameters {$\alpha_{(1,1)},\cdots, \alpha_{(2^D-1, K)}, \beta_{(1,1)}, \cdots, \beta_{(N,2^D)}$} are all positive.
> >
> > Assumption 5.2: The free parameters {$\gamma_{(1,1)},\cdots, \gamma_{(D, K)}, \beta_{(1,1)}, \cdots, \beta_{(N,2^D)}$} are all positive.
> >
> > In addition, in response to the example you proposed where an observed variable remains constant, we have reformulated the condition in the model setting to avoid the redundancy. Specifically, we assume the existence of a subspace $S(\mathbf{X})$ of the observed variables, where $S: \mathbb{R}^N \to \mathbb{R}^m $ is a mapping that selects or transforms a subset of the observed variables, such that $\mathrm{rank}(P(S(\mathbf{X}) \mid \mathbf{u})) = 2^D $ (For further clarification, we have updated the notation for “multi-domain” to a random variable “$\mathbf{u}$” in the main text).
> >
> > This condition ensures that the selected or transformed subspace $ S(\mathbf{X}) $ captures sufficient variability to fully encode the information from the $ D $ latent variables. Importantly, the rank condition guarantees that the configurations of the latent variables remain distinguishable through $ P(S(\mathbf{X}) \mid \mathbf{u}) $. For simplicity, in the main paper, we assume this subspace corresponds to the original variable space, with the number of variables still denoted as $N$.
> >
> > >***W2.4***: Assumption 6, which is core to the proof, is almost identical to the target proposition, making it seemingly circular
> >
> > - ***A2.4***: Thank you for your feedback. As the answer to your question above shows, we have modified Assumption 6 and put it as one of the general assumptions we listed in Assumption 1 to avoid any possible redundancy or circularity. We want to clarify that this rank assumption on $P(\mathbf{X}\mid \mathbf{u})$ does not necessarily imply identifiability (the proposition), as the assumption does not eliminate the possibility of existing another set of parameters where the rank constraint is satisfied and yields the same distribution of $ P(\mathbf{X})$.

---

> > > ### Author Response · Authors · 2024-11-24
> > >
> > > >***W3***: The paper claims in the abstract: 'In this paper, we show how multidomain information can be leveraged to achieve identifiability.' However, it's not clear from the paper how the multidomain setup "leveraged" identifiability. Moreover, given that identifiability is a model property, it's unclear how a 'multidomain' setup can be 'leveraged.' Furthermore, the need for a new work for the multi-domain setup isn't well explained, which is problematic given this is the paper's claimed novelty.
> > >
> > > - ***A3***: Thank you for your thoughtful comment. First, we would like to clarify that the multi-domain setup is also an inherent property of our model. Specifically, it means that the distributions of latent variables depend on an auxiliary variable as part of the model generation process. We leverage this property to prove identifiability.
> > > - The motivation for introducing multi-domain settings in our work is as follows: Most, if not all, previous studies on identifiability in multi-domain scenarios assume that the variables are continuous. In contrast, we are the first to establish identifiability in the discrete case and demonstrate how multi-domain information can significantly relax the parameter constraints that would otherwise be more restrictive in single-domain settings.
> > >
> > > - For example, assuming independent latent variables, when the number of latent variables is 3 $(D=3)$ and the number of observed variables is also 3 $(N=3)$, the distributions are not locally identifiable, as demonstrated by the simulation results in Table 1. However, if we have 6 or more domains, we can effectively identify all distributions, even when only 3 observed variables are available per domain.
> > >
> > > - In real-world applications, obtaining multiple domains is often practical and feasible. For instance, participants’ demographic information in survey data, user information in recommender systems, and object categories in image or video analysis are examples where multi-domain information is readily available and can be leveraged.
> > >
> > >
> > > >***W4***: There is insufficient explanation of how the provided experiments prove the claimed identifiability. While the paper derives sufficient conditions for identifiability, there is no discussion about whether these conditions are necessary and to what extent, thus limiting the paper's contribution.
> > >
> > > - ***A4***: Thank you for your feedback. We would like to clarify the purpose of the experiments in the paper, which include both simulation studies and real-world demonstrations of our identifiability results. In the simulation studies, we evaluated both local and strict identifiability by calculating the average KL divergence between the estimated and true distributions. When the KL divergence is less than $e^{-4}$, we consider the distributions identical and conclude that identifiability holds in such cases. For local identifiability, we demonstrated that our theoretical conditions are always sufficient (as the KL divergence consistently falls below the threshold) and, in certain cases, seem necessary, such as $N=5$  and  $D=3$  in Table 2. For the real-world demonstrations, we used two datasets: one inherently discrete (the Big Five dataset) and another derived from discretizing a continuous dataset (the NASDAQ-listed stocks dataset). In these examples, we showed that, given a sufficient number of domains, the estimated conditional independence relations between the latent and observed variables are meaningful and align with expectations. If you have any further questions about the experiments, we would be happy to provide additional clarification.
> > >
> > > - Regarding the identifiability conditions, it is worth noting that, in the existing literature, most identifiability conditions (if not all) are only sufficient (e.g., Hyvärinen \& Sasaki, 2019; Khemakhem et al., 2020). Deriving necessary conditions is extremely challenging, particularly in the complex and involved nonparametric setting.
> > >
> > > >***W5***: The paper's title focuses on discrete distributions, although the content appears to address only binary distributions.
> > >
> > > - ***A5***: Thank you for bringing up this question. For observed variables, as outlined in the Problem Statement section, categorical variables can always be transformed into binary form using standard encoding methods, such as one-hot encoding (see Bishop, 2006; Murphy, 2012). The same idea applies to latent variables. To recover potential latent variables that were binarized, a straightforward approach is to cluster binary variables that have strong effects on similar sets of observed variables and combine them into a single variable with multiple categories. While this is separate from the main focus of the paper, it is an area of future research we intend to explore.

---

> > > > ### Author Response · Authors · 2024-11-24
> > > >
> > > > >***W6***: Section 3 appears redundant as no other sections rely on it, and its results are trivial under such strict assumptions, leading to a degenerate model. The same applies to Theorem 2.
> > > >
> > > > - ***A6***:Thank you for your advice. We believe the reasoning here aligns with our responses to your earlier questions regarding Assumption 2. In summary, the one-to-one mapping case presented in this work is specifically designed as a counterpart to the continuous case (nonlinear ICA), which has been extensively studied in the literature, making it a valuable area of study.
> > > >
> > > > **References**
> > > >
> > > > [1]Hyvärinen, A., Karhunen, J., & Oja, E. (2001). Independent Component Analysis. John Wiley & Sons.
> > > >
> > > > [2]Hyvärinen, A., & Oja, E. (2000). Independent component analysis: Algorithms and applications. Neural Networks, 13(4-5), 411–430. https://doi.org/10.1016/S0893-6080(00)00026-5
> > > >
> > > > [3]Hyvärinen, A., & Morioka, H. (2017). Nonlinear ICA using temporal structure and smoothness. Proceedings of the 20th International Conference on Artificial Intelligence and Statistics (AISTATS), 54, 860–869. http://proceedings.mlr.press/v54/hyvarinen17a.html
> > > >
> > > > [4]Hyvärinen, A., & Sasaki, H. (2019). Nonlinear ICA using auxiliary variables and generalized contrastive learning. Proceedings of the 22nd International Conference on Artificial Intelligence and Statistics (AISTATS), 89, 859–868. http://proceedings.mlr.press/v89/hyvarinen19a.html
> > > >
> > > > [5]Khemakhem, I., Kingma, D. P., Monti, R., & Hyvärinen, A. (2020). Variational autoencoders and nonlinear ICA: A unifying framework. Proceedings of the 23rd International Conference on Artificial Intelligence and Statistics (AISTATS), 108, 2207–2217. http://proceedings.mlr.press/v108/khemakhem20a.html
> > > >
> > > > [6]Zheng, Y., Ng, I., & Zhang, K. (2022). On the identifiability of nonlinear ICA: Sparsity and beyond. Advances in Neural Information Processing Systems (NeurIPS), 35, 18957–18969. https://proceedings.neurips.cc/paper_files/paper/2022/hash/476ea3a0f3c4225f95c98ca7db67cfdb-Abstract-Conference.html

---

> > > > ### Author Response · Authors · 2024-11-24
> > > >
> > > > >***Q1***: Lines 036-038: Citation needed?
> > > >
> > > > ***A1***: Thanks for the suggestion, we have added more citations to the section (see Hyvärinen et al. (2001); Hyvärinen \& Morioka (2017)).
> > > >
> > > > >***Q2***: Lines 050-051, '...deep learning techniques to ensure identifiability': This requires clarification, as identifiability is a model property rather than an algorithmic one. How can deep neural networks 'ensure' identifiability?
> > > >
> > > > ***A2***: Thanks for the question. We have revised the manuscript to avoid such ambiguity: "For instance, when variables are continuous, it is well-known that the model becomes severely unidentifiable if the observations are independent and identically distributed (i.i.d.). To address this issue, additional constraints are typically required to ensure identifiability."

---

> > > > > ### Comment · Reviewer_X6ob · 2024-11-25
> > > > >
> > > > > I thank the authors for their thorough comments and for addressing most points raised. However, I believe the required changes are too vast for accepting the paper at this stage. Hence, I will keep my score.
> > > > > I hope a refined version of the paper will be published in a top venue in the future.

---

> > > > > > ### Author Response · Authors · 2024-11-27
> > > > > >
> > > > > > Thank you for your sincere and valuable suggestions on the paper; they have been very helpful to us. We have updated the manuscript accordingly. Please feel free to raise any questions or let us know if there are any aspects you find unsatisfactory.

---

### Meta-Review · Area_Chair_LDBW · 2024-12-22

**Metareview:**

Identifiability of discrete latent variables from discrete observations is studied. However, the reviewers consider the proofs unclear and not novel. Validity and practicality of the identifiability assumptions are also criticized. Theoretical identifiability is not corroborated by the numerical experiments. Moreover, all the real data experiments start by discretizing continuous data, which goes against the motivation of the proposed model.

**Additional Comments On Reviewer Discussion:**

Rebuttals are not able to convince the negative reviews.

---

### Decision · Program_Chairs · 2025-01-22

Reject